# *GFedCL*: Graph-Based Federated Continual Learning with Spatial and Temporal Awareness

Qingyang Yu [1]   Yang Hua [2]   Qizhen Zhang [3]   Hao Wang [1]

## Abstract

Recent years have witnessed a surge of interest in federated learning. In particular, federated continual learning (FCL) has emerged as an effective approach that enables clients with evolving, non-storable data to engage in collective learning. Among FCL approaches, replay-based methods excel by mitigating data storage constraints through synthetic data generation. However, existing replay-based methods overlook spatial and temporal information inherent in FCL settings, leading to suboptimal model performance. For instance, spatial variation in COVID-19 prevalence across hospitals in different states (*e.g.*, Delta surging in Florida vs. Omicron in New York) and the temporal evolution of regional outbreaks are critical information for accurately distinguishing between COVID variants. This paper presents GFedCL to address this limitation. GFedCL is a new FCL approach that (1) constructs spatial- and temporal-aware relational graphs with attention mechanisms, and (2) uses the graphs, combined with generative adversarial learning, to generate high-quality synthetic data. GFedCL generates synthetic data whose distributional expectation matches that of the real data distribution while preserving privacy with theoretical guarantees. GFedCL consistently outperforms state-of-the-art FCL methods, achieving 27.95% improvement on *TinyImageNet*.

## 1. Introduction

Conventional federated learning (FL) methods, such as FedAvg (McMahan et al., 2017), are widely used in domains, *e.g.*, healthcare, to collaboratively train models that generalize across all clients' data (tasks) while preserving privacy. These methods typically assume that each client's task is static, *i.e.*, the dataset remains fixed throughout the training process (Yang et al., 2025). However, in real-world scenarios, client data often evolves over time (Callaway, 2021). For instance, Figure 1 illustrates how influenza-like illness (ILI) distributions across four states in the United States change temporally (Centers for Disease Control and Prevention, 2025). Ignoring temporal dynamics confines the applications of FL to static tasks, limiting their generalization to tasks that vary over time periods in many practical applications (Wynants et al., 2020). To overcome this limitation, federated continual learning (FCL) (Yoon et al., 2021) incorporates principles from continual learning (CL) and offers a promising framework for training models that generalize across both clients and time in federated settings.

FCL has two primary challenges: *catastrophic forgetting* and *statistical heterogeneity*. Specifically, *catastrophic forgetting* refers to the degradation in model performance on previously learned tasks, which arises because clients are unable to retain data from earlier tasks due to various constraints, *e.g.*, privacy (Parisi et al., 2019) and limited storage (Parisi et al., 2019). *Statistical heterogeneity* refers to skewed, non-IID data distributions across clients (McMahan et al., 2017). To address these challenges, a rich literature of FCL has been proposed by exploring several orthogonal methodologies, *e.g.*, task-based methods (Yoon et al., 2021), regularization-based methods (Dong et al., 2022; Guo et al., 2021; Huang et al., 2022; Ma et al., 2022; Usmanova et al., 2021), replay-based methods (Abudukelimu et al., 2024; Sara et al., 2023; Qi et al., 2023), among others (He et al., 2025). In particular, replay-based methods proactively generate synthetic data that mimics previous tasks to refresh the model and thus can effectively mitigate *catastrophic forgetting* (van de Ven & Tolias, 2019).

Existing replay-based methods, unfortunately, utilize only generic information, such as class labels from previous tasks (Qi et al., 2023; Sara et al., 2023) or generative mod-

[1]Department of ECE, Stevens Institute of Technology, New Jersey, United States [2]Department of EEECS, Queen's University Belfast, Belfast, United Kingdom [3]Department of CS, University of Toronto, Ontario, Canada. Correspondence to: Qingyang Yu <qyu13@stevens.edu>.

*Proceedings of the 43rd International Conference on Machine Learning*, Seoul, South Korea. PMLR 306, 2026. Copyright 2026 by the author(s).

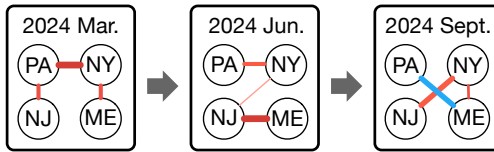

*Figure 1.* Temporal variation of influenza-like illness (ILI) distributions across four U.S. states. Each state is sequentially linked to the one with the most similar ILI pattern via edges. Edge color (Flu A or Flu B) indicates the dominant flu type, with thicker edges showing stronger dominance.

eling using normalizing flows (Abudukelimu et al., 2024). *They overlook the characteristics of FL clients and cannot utilize their spatial and temporal relationships over time.* Therefore, these approaches fall short in many real-world scenarios. For example, in the case of COVID-19, the same variant label, such as "Omicron," may be assigned to cases across different regions, even though the underlying feature distributions vary geographically (Chen et al., 2022). Similarly, data labeled "Omicron" in different time periods may reflect different feature patterns due to the continuous evolution of the virus (Viana et al., 2022). Ignoring such information leads to models biased toward specific manifestations of the variant, thereby limiting generalization. Furthermore, existing replay-based methods reactively mitigate statistical heterogeneity among clients, *e.g.*, using outlier feature removal (Abudukelimu et al., 2024), rather than proactively modeling it, and are thus less effective than dedicated distribution alignment strategies (Ye et al., 2022).

To leverage the crucial spatial and temporal information to efficiently mitigate *catastrophic forgetting* and *statistical heterogeneity*, we propose GFedCL, a novel replay-based FCL method that incorporates spatial and temporal relationships across clients and over time with privacy preservation.

*To model spatial and temporal relationships*, we first apply a spatial attention mechanism to construct a relational graph, where each edge encodes the pairwise attention score between clients' data distributions of the current task. We further employ a temporal attention mechanism to refine these scores by incorporating information from previous tasks. By combining the two attention mechanisms, GFedCL can capture both the relationships within the current task and those across longitudinal tasks.

*To simultaneously mitigate catastrophic forgetting and statistical heterogeneity*, we propose: (1) a purposely designed adversarial learning framework in which encoder–predictor pairs are initialized on each client, while a global discriminator is deployed on the server; and (2) a specialized objective for the global discriminator, which is trained to predict relational graphs rather than producing conventional "real/fake" labels as discriminators in standard adversarial learning. This design enables GFedCL to generate synthetic latent

representations, data used for task replay, that closely resemble those extracted from real data and align the data distribution of tasks across clients and over time. Catastrophic forgetting and statistical heterogeneity, the negative effects of inaccurate synthetic data in replay-based methods, can thus be mitigated.

*Additionally, to ensure privacy preservation during training*, we identify the unique potential attacks introduced by GFedCL compared to conventional FL methods, such as FedAvg. Then, we adopt effective security measures based on differential privacy (DP) to defend against attacks, validated by both empirical evidence and theoretical analysis. This design enables GFedCL to preserve privacy while maintaining sufficient model performance during training.

Overall, our contributions are as follows:

- We propose GFedCL, a novel FCL framework that exploits spatiotemporal dynamics inherent in FCL scenarios. By encoding this information in an auxiliary graph to guide the adversarial learning process, GFedCL aligns the expectations of the real and synthetic data distributions. Specifically, our theoretical analysis demonstrates that at the equilibrium of the underlying adversarial game, the expectation of the generated synthetic distribution converges to that of the real data distribution. This property allows GFedCL to effectively mitigate *catastrophic forgetting* and *statistical heterogeneity*.

- We enhance GFedCL's robustness with DP and prevent privacy leakage from attacks, such as membership inference attack (MIA) and distribution inference attack (DIA). Meanwhile, our theoretical analysis shows that the noise introduced by DP does not degrade the quality of the generated synthetic data at equilibrium.

- We present empirical results on synthetic and real-world datasets, showing that GFedCL significantly outperforms the state-of-the-art FCL methods. On *TinyImageNet* dataset, GFedCL achieves a 27.95% accuracy improvement compared to the best FCL baseline.

## 2. Related Work

**Centralized Continual Learning (CCL)** has witnessed the emergence of various methodologies (De Lange et al., 2022), which can be divided into three families: (1) Regularization-based methods (Li & Hoiem, 2016; Mirzadeh et al., 2020) employ a regularization term such as knowledge distillation to constrain the divergence between models of two consecutive tasks, assuming the distributions of consecutive tasks are similar; (2) Task-based methods (Rusu et al., 2016) leverage task IDs, a type of additional input, to explicitly indicate which tasks are being trained; (3) Replay-based

methods (Odena et al., 2017; Wu et al., 2018) proactively generate synthetic data with similar distributions to previous tasks to address catastrophic forgetting. All these methods have been extensively explored in centralized settings. However, due to the recent focus on the more complex and challenging problem setting, class-incremental learning (CIL), where task-specific information (*e.g.*, task IDs) is not provided and there is no class overlap between tasks, regularization-based and task-based methods have only achieved suboptimal performance (van de Ven & Tolias, 2019). In contrast, replay-based methods, which do not require task-specific information and make no assumptions about similarity between consecutive tasks, are promising in CIL, as they proactively generate synthetic images that are similar to real data from previous tasks.

**Federated Continual Learning (FCL)** aims to alleviate *catastrophic forgetting* and *statistical heterogeneity*. Specifically, $N$ clients $\{C_i\}_{i \in [N]}$ have their own tasks $\{\{T_i^k\}_{k \in [K]}\}_{i \in [N]}$ where $K$ is the number of tasks. The dataset for task $k$ of client $i$ is denoted as $\mathcal{D}_i^k = \{(x, y) \mid (x, y) \sim P_{T_i^k}\}$, representing the labeled samples drawn from the task distribution $P_{T_i^k}$. The training goal is to maximize model performance on downstream objectives, *e.g.*, classification accuracy over all tasks.

Previous studies in FCL can be categorized into several groups: (1) regularization-based methods (Dong et al., 2022; Huang et al., 2022; Ma et al., 2022; Usmanova et al., 2021), (2) task-based methods (Yoon et al., 2021), (3) replay-based methods (Abudukelimu et al., 2024; Sara et al., 2023; Qi et al., 2023), and others (Casado et al., 2020; Hendryx et al., 2021; He et al., 2025). Mirroring centralized continual learning (CCL), regularization and task-based methods underperform relative to replay-based approaches in class-incremental settings (Section 5), establishing the latter as the preferred choice for FCL. However, previous replay-based methods in FCL have only employed primitive replay strategies, such as using only class labels to guide the generation of synthetic data, neglecting the additional information introduced by FCL scenarios. FL introduces spatial relationships across clients, and the combination of FL and CL further involves the temporal relationships between clients over time. Thus, previous studies neglecting such spatial and temporal information may exhibit suboptimal performance in real-world scenarios (Section 5). In comparison, our proposal employs spatial and temporal attention to construct a relational graph, which guides synthetic data generation to mitigate *catastrophic forgetting* and *statistical heterogeneity*, enhancing the model performance in FCL scenarios.

## 3. Methods

### 3.1. GFedCL

Unlike prior replay-based FCL methods that rely on simplistic replay heuristics, GFedCL explicitly models the spatial and temporal structure inherent in FCL. Specifically, GFedCL employs a server-side attention mechanism that takes client classifier updates $\mathcal{Q}$ as input and constructs relational graphs $G \in \mathbb{R}_+^{N \times N}$ to capture inter-client and temporal dependencies. Additionally, to mitigate feedback-induced model collapse (Goodfellow et al., 2014), our relational graph construction is decoupled from training, preventing synthetic data from reinforcing spurious structures (Appendix C). Then, to leverage relational graphs for synthetic data generation while ensuring robust feature extraction, we adopt a dual-component generative adversarial learning framework. Specifically, we employ local *encoder-generator-predictor* triplets, $(E_i, \mathcal{G}_i, F_i)$, on each client $C_i$. Encoder, $E_i$, extracts latent representations from the current task $T_i^k$. Crucially, to prevent label leakage, $E_i$ takes as input only the features, $x \sim \mathcal{D}_i^k$, and the respective row of the relational graph, $\mathbf{g}_i^k \triangleq G_{i,:}^k \in \mathbb{R}_+^N$: $\mathcal{E}_{i,\text{real}}^k = E_i(x, \mathbf{g}_i^k)$. Generator, $\mathcal{G}_i$, generates synthetic latent representations for tasks $\{T_i^j\}_{j \in [k]}$. The input of $\mathcal{G}_i$ includes random noise $z \sim \mathcal{N}(0, I)$, the target label $y \sim \mathcal{D}_i^j$, and the row $\mathbf{g}_i^j$: $\mathcal{E}_{i,\text{syn}}^j = \mathcal{G}_i(z, y, \mathbf{g}_i^j)$. To enhance the discriminator's distinguishing capability and avoid the impact of FL aggregation, we further employ a global discriminator on the server that outputs predicted relational graphs. This design contrasts with initialize separate discriminators on clients (Appendices C and B).

We aim to learn a global encoder $E_g$, generator $\mathcal{G}_g$, and predictor $F_g$ such that extracted and generated features are invariant to the spatial and temporal dynamics encoded in the relational graphs. Thus, the distributions of real encodings (from $E_g$) and synthetic encodings (from $\mathcal{G}_g$) are aligned in a shared, task-agnostic latent space. The global predictor $F_g$ can hence accurately predict labels for both current and past tasks (via replay) without overfitting to client-specific contexts, essentially addressing *catastrophic forgetting* and *statistical heterogeneity*. Formally, given any input tuple $(x, \mathbf{g}_i^k)$ or $(z, y, \mathbf{g}_i^j)$, the global encoder $E_g$ extracts task-invariant feature representations $\mathcal{E}_{i,real}^k = E_g(x, \mathbf{g}_i^k)$ and the generator produces $\mathcal{E}_{i,syn}^k = \mathcal{G}_g(z, y, \mathbf{g}_i^j)$, such that the dependency between the embeddings and the graph is minimized: $\mathbb{E}[\mathbf{g}_i^k | \mathcal{E}_{i,\text{real}}^k] = \mathbb{E}[\mathbf{g}_i^k | \mathcal{E}_{i,\text{syn}}^k]$. This objective is achieved by leveraging a server-side global discriminator, while simultaneously training and aggregating local models from clients. The optimization in $C_i$ with task $T_i^k$ is:

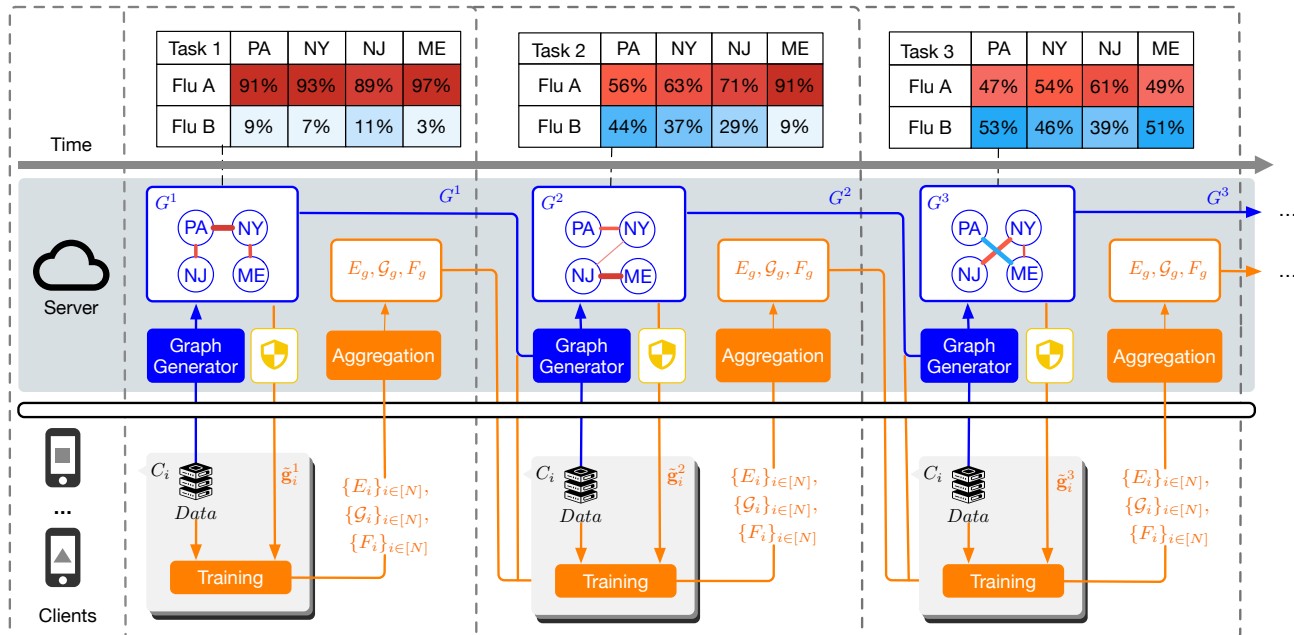

*Figure 2.* GFedCL's workflow on real-world tasks of flu diseases (Centers for Disease Control and Prevention, 2025). The intensity of colors in tables and graph edges of $G^1$, $G^2$, and $G^3$ reflects the level of dominance: the thicker the edge, the more dominant the corresponding flu type. Blue indicates the process of graph generation and orange indicates the training and aggregation.

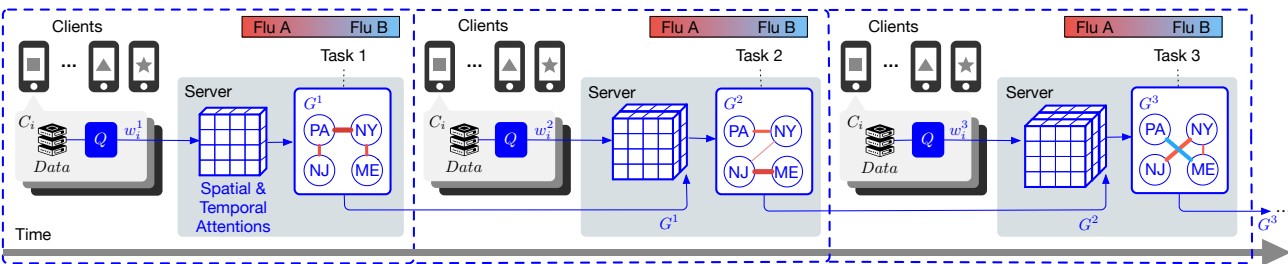

*Figure 3.* Graph generation integrates both spatial and temporal information. For task $T_i^k$, clients transmit local classifier updates $w_i^k$ to the server $S$. Subsequently, the server employs a spatio-temporal attention mechanism to compute scores $G_{ij}^k$ and construct the graph $G^k$.

$$\min_{E_i,\mathcal{G}_i,F_i} \max_D \underbrace{V_p^k(E_i, F_i) - \lambda_d V_d^k(D, E_i)}_{\text{Real Data (Current Task } k)} +$$
$$\underbrace{\sum_{j=1}^{k}(V_{p,\text{syn}}^j(\mathcal{G}_i, F_i) - \lambda_d V_{d,\text{syn}}^j(D, \mathcal{G}_i)))}_{\text{Synthetic Data (Current Task } k + \text{ Previous Tasks } 1\ldots k-1)} ,$$
$$(1)$$

where we have:

$$V_p^k(E_i, F_i) \triangleq \mathbb{E}_{(x,y)\sim\mathcal{D}_i^k}[\mathcal{L}_p(F_i(E_i(x, \mathbf{g}_i^k)), y)],$$

$$V_d^k(D, E_i) \triangleq \mathbb{E}_{(x,y)\sim\mathcal{D}_i^k}[\mathcal{L}_d(D(E_i(x, \mathbf{g}_i^k) + \tau_i), \mathbf{g}_i^k + \tau_{\mathbf{g}})],$$

$$V_{p,\text{syn}}^j(\mathcal{G}_i, F_i) \triangleq \mathbb{E}_{z\sim\mathcal{N}(0,I), y\sim\mathcal{D}_i^j}[L_p(F_i(\mathcal{G}_i(z, y, \mathbf{g}_i^j)), y)],$$

$$V_{d,\text{syn}}^j(D, \mathcal{G}_i) \triangleq$$
$$\mathbb{E}_{z\sim\mathcal{N}(0,I), y\sim\mathcal{D}_i^j}[L_d(D(\mathcal{G}_i(z, y, \mathbf{g}_i^j) + \tau_i), \mathbf{g}_i^j + \tau_{\mathbf{g}})],$$

where $\mathcal{L}_p$ is the prediction loss (*e.g.*, cross-entropy loss for classification tasks) and $\mathcal{L}_d$ is the discriminator loss. $\lambda_d$ is the hyperparameter to balance two different loss functions; however, under the realizability assumption, $\lambda_d$ does not alter the characterized equilibrium, only the convergence dynamics. $\tau_i$ and $\tau_{\mathbf{g}}$ are noises drawn from the Laplace distribution with mean $\mu = 0$ and scale $b = \frac{\Delta f}{\epsilon}$, where $\Delta f$ is the dataset sensitivity (commonly set to 1) and $\epsilon$ is the privacy budget.

### 3.2. Algorithm

GFedCL first initializes the local encoders $\{E_i\}_{i\in[N]}$, generators $\{\mathcal{G}_i\}_{i\in[N]}$, and predictors $\{F_i\}_{i\in[N]}$ from global states, a local classifier $\mathcal{Q}$ on each client, the global discrimina-

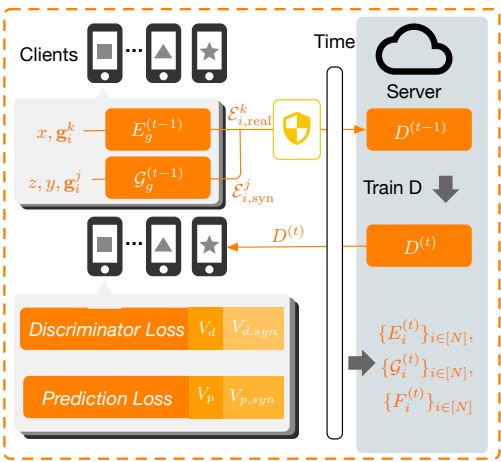

*Figure 4.* Training and aggregation in Round *t* with Task *k*.

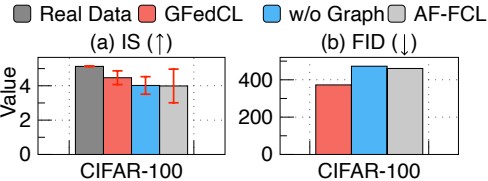

*Figure 5.* Leveraging spatio-temporal relational graphs, GFedCL outperforms SOTA on *CIFAR-100* IS and FID scores.

tor $D$ as $D^{(0)}$, and the spatial and temporal attentions on the server side. Then, GFedCL performs training via the following steps in Figure 2:[1]

**Step 1 (Graph Generation):** Figure 3 shows the process of graph generation: (1) Each client $C_i$ trains the local classifier $Q$ and uploads model updates to server $S$; (2) Server $S$ employs spatial and temporal (if $k > 1$) attentions to generate a relational graph of the current task $k$.

**Step 2 (Training):** Figure 4 shows the process of local training: (1) Each client $C_i$ uploads noised real encodings $\tilde{\mathcal{E}}_{i,\text{real}}^k = E_i(x, \mathbf{g}_i^k) + \tau_i$ and noised synthetic encodings $\tilde{\mathcal{E}}_{i,\text{syn}}^j = \mathcal{G}_i(z, y, \mathbf{g}_i^j) + \tau_i$ along with noised graphs to server $S$; (2) Server $S$ trains $D$ to minimize the prediction error of the graph $G$ (maximizing Eqn. 1 w.r.t $D$); (3) Server broadcasts updated $D$; (4) Each client trains $E_i, \mathcal{G}_i, F_i$ (minimizing Eqn. 1 w.r.t. $E_i, \mathcal{G}_i, F_i$).

**Step 3 (Aggregation):** Each client uploads $E_i, \mathcal{G}_i, F_i$ to server $S$ for aggregation. Then, server $S$ broadcasts updated global models to each client for the next round of training.

### 3.3. Spatio-Temporal Relational Graph

Analogous to conditional GANs (cGANs) (Mirza & Osindero, 2014) that utilize class labels or structural priors (Johnson et al., 2018) to guide generation, we condition our synthesis on a *relational graph*. This graph encodes the spatial and temporal alignment of data distributions across clients, effectively serving as a dynamic, continuous task identifier (van de Ven & Tolias, 2019).

Formally, we construct the relational graph matrix $G$ by synthesizing spatial correlations and temporal evolution via dual attention mechanisms. First, to capture *spatial* relationships within tasks $\{T_i^k\}_{i\in[N]}$, we compute the attention

---

[1]Appendix D presents the complete algorithm in pseudo-code.

score $\alpha_{ij}^k$ between clients $C_i$ and $C_j$ based on their current model updates (parameters) $w_i^k$ and $w_j^k$:

$$\alpha_{ij}^k = \text{softmax}_j \left( a(w_i^k, w_j^k) \right), \alpha_{ij}^k \in (0, 1),$$

where $a(\cdot)$ denotes the attention mechanism from (Veličković et al., 2018). Second, to model *temporal* dynamics (*e.g.*, concept drift or viral mutation patterns), we compute a temporal score $\beta_{ij}^k$ utilizing historical information $\{H_i^k\}_{i\in[N]}$ over a sliding window $m$:

$$\beta_{ij}^k = \text{softmax}_j \left( b(H_i^k, H_j^k, m) \right), \beta_{ij}^k \in (0, 1),$$

where $H_i^k$ represents the temporal variation pattern of client $C_i$, *i.e.*, concatenated $\{\mathbf{g}_i^{k-1}, \ldots, \mathbf{g}_i^{k-m-1}\}$ and $m$ is the sliding window length, and $b(\cdot)$ follows the temporal attention in (Shi et al., 2021).

The final relational graph edge weights are defined as the element-wise product:

$$G_{ij}^k = \alpha_{ij}^k \cdot \beta_{ij}^k \in (0, 1),$$

ensuring the graph is both spatially and temporally aware. This modular design allows for future extensions, such as incorporating topological metrics such as betweenness centrality (Freeman, 1977) into the spatial attention score.

As illustrated in Figure 5, this structural conditioning improves generation quality, *e.g.*, inception score (IS) and Fréchet inception distance (FID), compared to state-of-the-art baselines, *e.g.*, AF-FCL (Abudukelimu et al., 2024).

### 3.4. Privacy

GFedCL differs from conventional FL (*e.g.*, FedAvg) by incorporating a global discriminator on the server, which processes data encodings and relational graphs for further training. However, this approach may expose sensitive information through data encodings and relational graphs. We should tackle potential private information leakage introduced by the additional data transmission in GFedCL, where attackers may attempt model inversion attacks such as DIA (Shokri et al., 2017) and MIA (Geiping et al., 2020). To defend against such attacks, we employ DP (Dwork et al., 2006) by adding varying Laplace noises to transmitted data encodings and fixed Laplace noise to transmitted relational

graphs in each communication round.[2] This distinction arises from the static structure of relational graphs, which remains constant during a task, in contrast to data encodings that vary dynamically with changes in the encoders across communication rounds. Other potential privacy concerns, such as gradient inversion, which have been extensively studied in prior research, *e.g.*, FLTrust (Cao et al., 2021), are beyond the scope of this work and not discussed here. Besides, safeguards to other potential privacy issues are orthogonal to GFedCL and can be easily incorporated (Appendix C).

## 4. Theoretical Analysis

Previous studies on adversarial learning using latent space representations (Donahue et al., 2017; Zhao et al., 2019) typically analyze the behavior of components (*e.g.*, $E$, $\mathcal{G}$, and $D$) at equilibrium, uncovering important properties of components and latent representations under optimal conditions. For example, BiGAN (Donahue et al., 2017) demonstrates that, at equilibrium, the optimal encoder is the reverse of the optimal generator, indicating that the synthetic data is indistinguishable from real ones.

Similarly, we show that GFedCL could achieve an important property: the synthetic data encodings $\mathcal{E}_{i,\text{syn}}$ (generated from random noise $z$) become identical to the real data encodings $\mathcal{E}_{i,\text{real}}$ (generated from real data $x$) at equilibrium. [3] This property highlights the quality of the synthetic data and further illustrates that the data distributions across tasks are well aligned when the system reaches its optimum. Specifically, we demonstrate this property in two cases:

- **Simplified GFedCL**: This scenario involves only three types of components, the encoders, the generator, and the discriminator. The analysis, in this case, focuses on the behavior of the optimal discriminator, encoders, and generators at equilibrium.

- **GFedCL**: In practical applications, the ultimate goal is not just to generate synthetic data identical to real data but also to ensure that the model performs well on the downstream task (*e.g.*, classification). Thus, this analysis considers the interaction among four components: encoder, generator, predictor, and discriminator, demonstrating how synthetic data generation, alignment, and prediction achieve equilibrium.

### 4.1. Analysis of the Simplified GFedCL

For the simplified GFedCL, each local client consists of an encoder $E_i$ and a generator $\mathcal{G}_i$ (without a predictor $F_i$). In this setting, the training process is modeled as a minimax

game. The global discriminator $D$ aims to reconstruct the local relational attention vector $\mathbf{g}$ from the embeddings to minimize the $L_2$ loss. Conversely, the encoder $E_i$ and generator $\mathcal{G}_i$ aim to maximize this loss to generate privacy-preserving, structurally invariant representations.

The joint optimization problem for client $C_i$ is defined as:

$$\max_{E_i, \mathcal{G}_i} \min_D \mathcal{L}(D, E_i, \mathcal{G}_i)$$

where the total objective $\mathcal{L}$ aggregates the reconstruction risks over both real and synthetic distributions:

$$\mathcal{L} = \underbrace{\mathbb{E}_{x \sim P_{T_i^k}}[||D(E_i(x, \mathbf{g}_i^k) + \tau_{\mathcal{E}}) - (\mathbf{g}_i^k + \tau_{\mathbf{g}})||_2^2]}_{\text{Real Data (Current Task } k)}$$

$$+ \sum_{j=1}^k \underbrace{\mathbb{E}_{z \sim P_z, y \sim P_{T_i^j}}[||D(\mathcal{G}_i(z, y, \mathbf{g}_i^j) + \tau_{\mathcal{E}}) - (\mathbf{g}_i^j + \tau_{\mathbf{g}})||_2^2]}_{\text{Synthetic Data (Current Task } k + \text{Previous Tasks } 1 \ldots k-1)}.$$

We first analyze the equilibrium behavior of the discriminator.

**Lemma 4.1** (Optimal Discriminator for GFedCL). *For fixed generative components $(E_i, \mathcal{G}_i)$, the optimal discriminator $D^*$ that minimizes MSE is given by the conditional expectation of the graph vector $\mathbf{g}$ given the noisy embedding[4] $\tilde{\mathcal{E}}$:*

$$D^*(\tilde{\mathcal{E}}) = \mathbb{E}_{\mathbf{g} \sim p(\mathbf{g}|\tilde{\mathcal{E}})}[\mathbf{g}],$$

*where $\mathbf{g} \in \mathbb{R}_+^N$ denotes the specific row of the relational graph matrix $G$ corresponding to the client and $\tilde{\mathcal{E}} = \mathcal{E} + \tau_{\mathcal{E}}$. This form holds universally for inputs $\tilde{\mathcal{E}}$ from both the real distribution, $P_{\tilde{\mathcal{E}}_{real}}$, and the synthetic distribution $P_{\tilde{\mathcal{E}}_{syn}}$.*

Given the optimal discriminator $D^*$, the generative components $(E_i, \mathcal{G}_i)$ seek to maximize the irreducible error. By the Law of Total Variance, this minimal loss equates to the expected conditional variance $\mathbb{E}_{\tilde{\mathcal{E}}}[\text{Tr}(\mathbb{V}[\mathbf{g}|\tilde{\mathcal{E}}])]$.

**Theorem 4.2** (Global Optimum for Simplified GFedCL). *The simplified GFedCL achieves the global optimum if and only if both the encoder $E_i$ and generator $\mathcal{G}_i$ produce encodings such that the conditional expectation of the graph structure is identical to the marginal expectation:*

$$\mathbb{E}[\mathbf{g}|\tilde{\mathcal{E}}_{real}] = \mathbb{E}[\mathbf{g}] \quad and \quad \mathbb{E}[\mathbf{g}|\tilde{\mathcal{E}}_{syn}] = \mathbb{E}[\mathbf{g}].$$

This result implies that the proposed minimax game forces the system to align distributions based on shared structural properties. We summarize the implications for heterogeneity and forgetting in the following corollaries.

---

[2]Appendix D outlines DP algorithm in pseudo-code.
[3]Appendix A.4 provides convergence analysis of GFedCL.

[4]For readability and simplicity, since such a fundamental mathematical property of MSE loss is universal, we keep notations in a generic form, *i.e.*, $\mathbf{g}$, $\tilde{\mathcal{E}}_{\text{syn}}$, $\tilde{\mathcal{E}}_{\text{real}}$, etc.

**Corollary 4.3** (Mitigation of Statistical Heterogeneity). *At the global optimum, $\tilde{\mathcal{E}}_{real} \perp\!\!\!\perp \mathbf{g}$, i.e., $\mathbb{E}[\mathbf{g} \mid \tilde{\mathcal{E}}_{real}] = \mathbb{E}[\mathbf{g}]$. This first-moment invariance implies that $E_i$ suppresses graph-induced bias at the level of conditional means, thereby reducing statistical heterogeneity across clients.*

**Corollary 4.4** (Structural Consistency). *Since real and synthetic embeddings satisfy the same first-moment invariance condition, i.e., $\mathbb{E}[\mathbf{g} \mid \tilde{\mathcal{E}}] = \mathbb{E}[\mathbf{g}]$, $\mathcal{G}_i$ achieves structural consistency with $E_i$, enabling generative replay to preserve latent-space structure and mitigate catastrophic forgetting.*

*Remark* 4.5 (Robustness to DP Noise). Additive Laplace noise $\tau_{\mathcal{E}}$ adds constant variance but does not change the equilibrium. If the embedding is first-moment graph-invariant, *i.e.*, $\mathcal{E} \perp\!\!\!\perp \mathbf{g}$, then this invariance is preserved after adding independent DP noise.

Theorem 4.2 shows that GFedCL achieves first-moment alignment, *i.e.*, mean alignment, between synthetic and real data embeddings in the latent space. Such first-moment alignment contributes to mitigating the two core challenges in FCL from two aspects: (1) the alignment among data embeddings of all clients' current tasks, *i.e.*, $\{\mathcal{D}_i^k\}_{i \in [N]}$, indicates that GFedCL reduces client-level distributional discrepancies in the latent space, thereby alleviating statistical heterogeneity across clients; and (2) the alignment between synthetic embeddings and real data embeddings indicates that the generated samples preserve the latent statistics of previously observed data, making replay more faithful to past knowledge and thus mitigating catastrophic forgetting. Therefore, the theoretical result supports the central mechanism of GFedCL: by organizing real and synthetic data from different tasks and clients into a shared latent representation space, GFedCL simultaneously improves knowledge retention and cross-client consistency.

The current formulation of GFedCL mainly enforces mean alignment because the discriminator is trained with an $L_2$ loss. Under this loss, the optimal discriminator provides a point estimate of the target distribution, *i.e.*, its conditional mean. Consequently, the resulting alignment objective naturally reduces to first-moment alignment in the latent space. To obtain high-order distributional alignment, we further introduce a probabilistic discriminator objective that aligns both the first and second moments, *i.e.*, the mean and variance, of synthetic and real data embeddings. This formulation remains general and can be extended to richer probabilistic objectives for higher-order alignment.

If the discriminator is instead designed to predict a distribution, then the alignment can naturally be extended from point-level to distribution-level matching. For example, one can use a Gaussian NLL loss and let the discriminator output both the mean and variance of the graph-conditioned latent distribution. In this case, the objective would align

not only the first moment but also the second moment:

$$\mathcal{L}_d = \frac{(D_\mu(\mathcal{E}_{i,}) - (\mathbf{g}_i + \tau_\mathbf{g}))^2}{2D_{\sigma^2}(\mathcal{E}_{i,})} + \frac{1}{2}\log D_{\sigma^2}(\mathcal{E}_{i,}).$$

Based on the above theoretical analysis, we have the following implications for the alignment.

*Remark* 4.6 (Alignment of High-Order Moments). Theorem 4.2 shows that GFedCL achieves the first moment (mean) alignment. Second-order moments (*e.g.*, mean and variance) are achievable via changing the discriminator's loss function (*e.g.*, from MSE to Gaussian NLL). High-order moments, *e.g.*, skewness, are also achievable via using skew-normal distribution.

### 4.2. Global Optimum of GFedCL

The equilibrium analysis for the complete GFedCL framework involves the four-player game between the encoder $E_i$, generator $\mathcal{G}_i$, predictor $F_i$, and discriminator $D$. We require the system to achieve not just distributional alignment (structural invariance), but also semantic correctness (accurate label prediction).

**Theorem 4.7.** *If the encoder $E_i$, generator $\mathcal{G}_i$, predictor $F_i$, and discriminator $D$ are trained to reach the global optimum, the system satisfies the following properties:*

1. ***Structural Alignment:*** *The expectations of marginal distributions of real and synthetic embeddings are identical: $\mathbb{E}[P(\mathcal{E}_{real})] = \mathbb{E}[P(\mathcal{E}_{syn})]$.*

2. ***Info Preservation:*** *$E_i$ retains all information about the label $y$ contained in the input $x$ and graph $\mathbf{g}$ under realizability with enough capacity, i.e., $H(y|\mathcal{E}_{real}) = H(y|x, \mathbf{g})$.*

3. ***Synthetic Semantics:*** *$\mathcal{G}_i$ produces features containing precise label information, i.e., $H(y|\mathcal{E}_{syn}) = 0$.*

Theorem 4.7 establishes that the optimal encoder retains all the information related to the label $y$ present in data $x$ while simultaneously achieving alignment with the synthetic data distributions at equilibrium.

While Theorem 4.2 and Theorem 4.7 characterize the encoder, predictor, generator, and discriminator at their optimum, due to the non-convex nature of the optimization, this optimum might never be reached. Empirically, Section 5 shows that on realistic datasets, GFedCL achieves improvement in model performance compared to baselines.

## 5. Experiments

We empirically validate the results of GFedCL on three synthetic datasets and two real-world medical datasets. Ad-

*Table 1.* Performance of GFedCL on training datasets (best result in **bold**, second-best underlined). For classification tasks on *EMNIST-Letters*, *CIFAR-100*, and *TinyImageNet*, we report *Accuracy* (%) and *Forgetting* (%). For the regression task on *US-States*, we report *MSE* loss only. Here, FedAvg-Adv refers to FedAvg-Adversarial. Due to method-design restrictions, $C^2$Prompt cannot be directly applied to the regression task on *US-States* and we therefore use "-" to denote the unavailable results.

| Method | EMNIST-Letters | | CIFAR-100 | | TinyImageNet | | TCGA-BRCA | | US-States |
|---|---|---|---|---|---|---|---|---|---|
| | Acc (%)↑ | Forgetting (%)↓ | Acc (%)↑ | Forgetting (%)↓ | Acc (%)↑ | Forgetting (%)↓ | Acc (%)↑ | Forgetting (%)↓ | MSE ($\times 10^{-2}$) ↓ |
| GLFC | $43.47 \pm 2.14$ | $8.74 \pm 2.78$ | $46.08 \pm 2.70$ | $5.66 \pm 1.74$ | $42.93 \pm 2.88$ | $7.64 \pm 2.91$ | $59.74 \pm 1.97$ | $5.08 \pm 1.17$ | $3.40 \pm 0.22$ |
| FedAvg-Adv | $25.67 \pm 4.49$ | $9.77 \pm 3.17$ | $32.05 \pm 4.51$ | $8.92 \pm 2.01$ | $25.41 \pm 6.55$ | $9.74 \pm 3.29$ | $32.70 \pm 5.20$ | $7.92 \pm 4.61$ | $3.95 \pm 0.20$ |
| FedCIL | $38.95 \pm 4.53$ | $8.87 \pm 2.31$ | $39.47 \pm 3.24$ | $6.54 \pm 1.72$ | $45.75 \pm 5.26$ | $7.51 \pm 1.33$ | $56.59 \pm 4.75$ | $8.13 \pm 2.40$ | $3.14 \pm 0.31$ |
| MFCL | $52.66 \pm 3.24$ | $6.76 \pm 1.24$ | $37.09 \pm 3.76$ | $5.26 \pm 1.87$ | $47.82 \pm 3.09$ | $8.11 \pm 1.07$ | $54.81 \pm 2.89$ | $8.24 \pm 1.84$ | $2.74 \pm 0.24$ |
| AF-FCL | $41.22 \pm 4.51$ | $6.58 \pm 1.44$ | $43.27 \pm 4.77$ | $5.87 \pm 2.35$ | $54.91 \pm 3.72$ | $7.84 \pm 2.02$ | $67.44 \pm 2.59$ | $\underline{2.92} \pm 1.04$ | $3.84 \pm 0.19$ |
| $C^2$Prompt | $68.47 \pm 3.81$ | $7.42 \pm 1.80$ | $\underline{67.93} \pm 5.19$ | $6.94 \pm 2.77$ | $\underline{72.41} \pm 4.53$ | $6.72 \pm 1.84$ | $\underline{72.48} \pm 3.11$ | $4.28 \pm 1.95$ | – |
| AFL | $\underline{75.17} \pm 1.31$ | $\mathbf{2.94} \pm 0.79$ | $55.33 \pm 1.55$ | $\mathbf{3.47} \pm 0.85$ | $49.44 \pm 1.35$ | $\mathbf{3.95} \pm 1.59$ | $68.48 \pm 1.49$ | $5.91 \pm 0.76$ | $\underline{2.62} \pm 0.11$ |
| GFedCL | $\mathbf{77.23} \pm 2.76$ | $\underline{3.34} \pm 1.21$ | $\mathbf{73.67} \pm 3.42$ | $\underline{4.24} \pm 1.45$ | $\mathbf{77.39} \pm 3.02$ | $\underline{5.57} \pm 1.88$ | $\mathbf{75.32} \pm 2.19$ | $3.93 \pm 1.25$ | $\mathbf{2.41} \pm 0.18$ |

ditional results, *e.g.*, ablation studies, sensitivity tests, and communication and computational overhead evaluation, are provided in Appendix B. These empirical results verify our theoretical analysis in Section 4 and show that GFedCL achieves:

- **Superior Performance:** GFedCL outperforms SOTA baselines, notably improving accuracy on *TinyImageNet* by 27.95% (Table 1).

- **Overhead Efficiency:** GFedCL incurs negligible communication and computational overhead compared to standard FCL methods (Appendices B.6–B.7).

- **Privacy & Robustness:** DP mechanisms within GFedCL ensure secure relational graph construction and robust data encoding (Appendix B.8).

## 5.1. Experiment Setups

**Datasets:** We select three benchmark datasets, *EMNIST-Letters* (Cohen et al., 2017), *CIFAR-100* (Krizhevsky, 2009), and *TinyImageNet* (Le, 2015), as well as two real-world medical datasets, *US-States* (Centers for Disease Control and Prevention, 2025) and *TCGA-BRCA* (The Cancer Genome Atlas Network, 2012).

**Baselines:** We include the following FCL methods as baselines: FedAvg-Adversarial (FedAvg-Adv) refers to a combination of a conditional generative adversarial network (GAN) (Mirza & Osindero, 2014) augmented with a predictor, integrated within the FedAvg framework. FedCIL (Qi et al., 2023) employs ACGAN (Odena et al., 2017) on each client and model consolidation in the server. MFCL (Sara et al., 2023) deploys a knowledge distillation model in the server to generate synthetic samples. AF-FCL (Abudukelimu et al., 2024) maps features extracted by ResNet (He et al., 2016) into different data distributions to generate synthetic samples via a normalizing flow model. GLFC (Dong et al., 2022) utilizes a regularization term to constrain the divergence of each client in consecutive tasks. AFL (He et al., 2025) models the feature-label relationships linearly,

*Table 2.* Ablation study on *US-States*. *Temp.* and *Attn.* denote Temporal and Attention modules, respectively. Full ablation studies on other components and datasets are in Appendix B.

| Dataset | GFedCL | w/o Temp. Attn. | w/o Graph |
|---|---|---|---|
| *US-States* | $2.41 \pm 0.18$ | $2.48 \pm 0.19$ | $2.52 \pm 0.24$ |

proposing a closed-form solution to FCL. $C^2$Prompt (Xu et al., 2025) employs stored historical class information to match new tasks.

**Evaluation Metrics:** We use *average accuracy* and *forgetting* (Mirzadeh et al., 2021) to evaluate classification on *EMNIST-Letters*, *CIFAR-100*, *TinyImageNet*, and *TCGA-BRCA*, and *MSE loss* to evaluate regression on *US-States*. Formal definitions of *average accuracy* and *forgetting* are in Appendix B.1.

**Configurations:** We follow a standard CIL setting (van de Ven & Tolias, 2019), *i.e.*, no class overlap between tasks of each client. Specifically, each client would randomly select $C$ classes from the training datasets for each task. For detailed setup, *e.g.*, number of clients ($N$), tasks per client ($K$), and classes per task ($C$), please refer to Appendix E.

## 5.2. Results on *EMNIST-Letters*

We begin with the common *EMNIST-Letters* dataset to evaluate GFedCL and the baselines. *EMNIST-Letters* comprises 26 classes of letters from "A" to "Z." Each class contains around 4800 training samples and 800 test samples. Each image is sized at 28 pixels × 28 pixels.

Table 1 shows that all prior replay-based methods, as well as the regularization-based method GLFC, provide suboptimal performance. In comparison, GFedCL achieves a 2.06% improvement in accuracy over the best-performing baseline (AFL) with comparable performance on forgetting. This result demonstrates the significant benefits of modeling both spatial and temporal relationships in the proposed relational graph for FCL.

## 5.3. Results on *CIFAR-100*

*CIFAR-100* includes 100 fine-grained classes that are grouped into 20 coarse (super) classes, each containing 5 fine classes. Each class contains 500 training images and 100 test images. Each $32\times32$ image has both a "fine" label (one of the 100 specific classes) and a "coarse" label (one of the 20 superclasses). In our experiments, we use the "fine" labels for both training and testing.

As shown in Table 1, this dataset represents a challenging scenario for existing solutions: their accuracies, including that of the second-best approach (AFL), are universally below 60%. Compared to these approaches, *GFedCL achieves a substantial 18.34% improvement.*

## 5.4. Results on *TinyImageNet*

*TinyImageNet* has 200 classes, which are based on the Word-Net (Miller, 1995) lexical database. Many of the classes can be broadly categorized based on semantic similarity or taxonomic groups, *e.g.*, animals, vehicles, *etc*. Each class contains 500 training samples and 50 validation samples.

Table 1 shows that this dataset is also demanding for the baselines, and AFL performs worse than AF-FCL (49.44% vs. 54.91%). While no existing approach can achieve accuracy higher than 55%, GFedCL is highly accurate for this task, achieving 77.39% accuracy—a 27.95% improvement.

## 5.5. Results on *TCGA-BRCA*

*TCGA-BRCA* is a real-world dataset that contains 1,095 samples, where each sample includes: (1) over 18,000 high-dimensional gene expression features (RNA-Seq), (2) 39 clinical features (*e.g.*, tumor stage, survival time, receptor statuses, etc.), (3) binary subtype labels distinguishing Basal-like vs. non-Basal tumors, and (4) demographic and clinical covariates (*e.g.*, age).

Table 1 shows that GFedCL again achieves better prediction accuracy than all the baselines: 75.32% vs. 68.48% (AFL, the existing best-performing approach).

## 5.6. Results on *US-States*

*US-States* is a real-world dataset, comprising 360 samples where each sample represents the number of weekly average ILI cases in forty-nine states. Typically, each sample is a vector of size $1\times49$, and each element is a digit. The samples are collected from 2010 to 2017. We divide the 49 states into seven groups (clients) based on geographic connectivity and define each task as a sequence of 60 consecutive samples, comprising 50 training and 10 test samples. The demonstration and visualization of group division are included in Appendix B.9.

Table 1 shows that GFedCL also performs better than other solutions on this task: its MSE (2.41) is consistently lower than that of AFL (2.62) and other baselines.

## 5.7. Ablation Study

We performed an ablation study to investigate the impact of integrating temporal and spatial information on the performance of GFedCL. Specifically, we first remove the temporal attention mechanism from the relational graph and then completely remove the relational graph from GFedCL and see how its performance differs. The results on *US-States* are shown in Table 2, which demonstrates that both mechanisms can improve GFedCL's performance.

## 6. Conclusion

We modeled the spatial and temporal structure of FCL using relational graphs and integrated them into GFedCL to address catastrophic forgetting and statistical heterogeneity. We provided both theoretical analysis and empirical results on a variety of datasets, demonstrating the feasibility of our proposal. These results highlight the practical potential of GFedCL in a wide range of application scenarios, *e.g.*, medical domains.

## Acknowledgment

We thank the anonymous reviewers for their valuable feedback. The work of Q. Yu and H. Wang was supported in part by the United States National Science Foundation (NSF) under grants 2523997, 2315612, and 2332638 and by the AWS Cloud Credit for Research program. The work of Q. Zhang was supported in part by the Natural Sciences and Engineering Research Council of Canada ALLRP 588144 - 23. Any opinions, findings, and conclusions or recommendations expressed in this material are those of the authors and do not necessarily reflect the views of the funding agencies.

## Impact Statement

This paper presents work whose goal is to advance the field of Machine Learning. There are many potential societal consequences of our work, none of which we feel must be specifically highlighted here.

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

## A. Theoretical Analysis

### A.1. Effectiveness of Differential Privacy

A formal definition of Local DP is provided in Definition A.1. The privacy parameter $\epsilon$ captures the privacy loss incurred by the algorithm's output. Specifically, $\epsilon \to 0$ ensures perfect privacy in which the output is independent of its input, while $\epsilon \to \infty$ offers no privacy guarantee.

**Definition A.1** (Differential Privacy). Let $A : I \to O$ be a randomized algorithm mapping a data entry in $I$ to $O$. The algorithm A is $(\epsilon, \delta)$-local differentially private if for all data entries $x, x' \in I$ and outputs $o \in O$, we have:

$$Pr\{A(x) = o\} \leq exp(\epsilon)Pr\{(A(x') = o\} + \delta. \quad (2)$$

Algorithm 2 outlines how to add noise to inputs. The parameters $\Delta f$ and $\epsilon$ correspondingly are the dataset sensitivity and privacy budget. Measuring the true sensitivity of representation is challenging. Hence, we follow previous work bounding the sensitivity with 1 (*i.e.*, $\Delta f = 1$) (Shokri & Shmatikov, 2015). Theorem A.2 proposes a formal statement and proof of Algorithm 2's privacy guarantees.

**Theorem A.2** (Local Differential Privacy). *Let the noise vector $\tau$ have i.i.d. entries drawn from* $\mathrm{Lap}(b)$ *with* $b = \frac{\Delta f}{\epsilon}$. *Then Algorithm 2 is $\epsilon$-locally differentially private.*

*Proof.* Let $f$ be a (possibly vector-valued) function mapping a dataset $I$ to $\mathbb{R}^d$. The $\ell_1$-sensitivity of $f$ is defined as

$$\Delta f = \max_{I \sim I'} \|f(I) - f(I')\|_1,$$

where $I$ and $I'$ differ in at most one entry.

Algorithm 2 outputs

$$A(I) = f(I) + \tau,$$

where each coordinate of $\tau$ is drawn independently from $\mathrm{Lap}(b)$. Thus, for any $y \in \mathbb{R}^d$, the probability density is

$$\Pr\{A(I) = y\} = \prod_i \frac{1}{2b} \exp\left(-\frac{|y_i - f(I)_i|}{b}\right).$$

Similarly,

$$\Pr\{A(I') = y\} = \prod_i \frac{1}{2b} \exp\left(-\frac{|y_i - f(I')_i|}{b}\right).$$

Taking the likelihood ratio yields

$$\frac{\Pr\{A(I) = y\}}{\Pr\{A(I') = y\}} = \prod_i \exp\left(\frac{|y_i - f(I')_i| - |y_i - f(I)_i|}{b}\right)$$

$$\leq \prod_i \exp\left(\frac{|f(I)_i - f(I')_i|}{b}\right)$$

$$= \exp\left(\frac{\sum_i |f(I)_i - f(I')_i|}{b}\right)$$

$$= \exp\left(\frac{\|f(I) - f(I')\|_1}{b}\right)$$

$$\leq \exp\left(\frac{\Delta f}{b}\right) = e^\epsilon,$$

where the last inequality follows from the definition of $\Delta f$ and $b = \frac{\Delta f}{\epsilon}$.

Therefore, Algorithm 2 satisfies $\epsilon$-local differential privacy.
□

Now, we prove the local DP of each client. To demonstrate the global DP of Algorithm 1, Theorem A.3 provides a formal statement for Algorithm 1's privacy guarantees.

**Theorem A.3** (Global Differential Privacy). *Let clients employ Algorithm 2 during training. Then, Algorithm 1 is globally differential private*

*Proof.* Given Algorithms 1 and 2, each client uploads and downloads data independently. Hence, global differential privacy is maintained during upload and download procedures. The discriminator in the server processes the uploaded data individually and independently, ensuring that differential privacy is not compromised. Therefore, the global differential of Algorithm 1 is achieved. □

### A.2. Global Optimum of Simplified GFedCL

This subsection provides a theoretical analysis for the simplified GFedCL, where each local client consists of an encoder $E_i$ and a generator $\mathcal{G}_i$ (without a predictor $F_i$). In this setting, the training process is modeled as a minimax game. The global discriminator $D$ aims to reconstruct the local relational attention vector **g** from the embeddings to minimize the $L_2$ loss. Conversely, the encoder $E_i$ and generator $\mathcal{G}_i$ aim to maximize this loss to generate privacy-preserving, structurally invariant representations.

The joint optimization problem for client $C_i$ is defined as:

$$\max_{E_i, \mathcal{G}_i} \min_D \mathcal{L}(D, E_i, \mathcal{G}_i), \quad (3)$$

where the total objective $\mathcal{L}$ aggregates the reconstruction

risks over both real and synthetic distributions:

$$\mathcal{L} = \underbrace{\mathbb{E}_{x \sim P_{T_i^k}}[||D(E_i(x, \mathbf{g}_i^k) + \tau_\mathcal{E}) - (\mathbf{g}_i^k + \tau_\mathbf{g})||_2^2]}_{\text{Real Data (Current Task } k)}$$

$$+ \sum_{j=1}^{k} \underbrace{\mathbb{E}_{z \sim P_z, y \sim P_{T_i^j}}[||D(\mathcal{G}_i(z, y, \mathbf{g}_i^j) + \tau_\mathcal{E}) - (\mathbf{g}_i^j + \tau_\mathbf{g})||_2^2]}_{\text{Synthetic Data (Current Task } k + \text{Previous Tasks } 1...k-1)}.$$

We first analyze the equilibrium behavior of the discriminator.[5]

**Lemma A.4** (Optimal Discriminator for GFedCL). *For fixed generative components* $(E_i, \mathcal{G}_i)$, *the optimal discriminator* $D^*$ *that minimizes the mean squared error is given by the conditional expectation of the graph vector* $\mathbf{g}$ *given the noisy embedding* $\tilde{\mathcal{E}}$:

$$D^*(\tilde{\mathcal{E}}) = \mathbb{E}_{\mathbf{g} \sim p(\mathbf{g}|\tilde{\mathcal{E}})}[\mathbf{g}],$$

*where* $\mathbf{g} \in \mathbb{R}_+^N$ *denotes the specific row of the relational graph matrix* $G$ *corresponding to the client. This form holds universally for inputs* $\tilde{\mathcal{E}}$ *from both the real distribution,* $P_{\tilde{\mathcal{E}}_{real}}$, *and the synthetic distribution* $P_{\tilde{\mathcal{E}}_{syn}}$.[6]

*Proof.* The objective function separates into independent terms based on the input source. For any input distribution $p(\tilde{\mathcal{E}}, \mathbf{g})$, the MSE is minimized by the conditional mean $D^*(\tilde{\mathcal{E}}) = \mathbb{E}[\mathbf{g}|\tilde{\mathcal{E}}]$ due to the standard bias-variance decomposition. Thus, $D^*$ learns to estimate the expected graph structure associated with any given embedding. □

Assuming $D$ achieves this optimum $D^*$, we analyze the objective for the generative components. Both $E_i$ and $\mathcal{G}_i$ seek to maximize the minimal loss achievable by $D^*$. By the Law of Total Variance, the minimal loss (the irreducible error) for a distribution is the expected conditional variance $\mathbb{E}_{\tilde{\mathcal{E}}}[\text{Tr}(\mathbb{V}[\mathbf{g}|\tilde{\mathcal{E}}])]$.

**Theorem A.5** (Global Optimum for Simplified GFedCL). *The simplified GFedCL achieves the global optimum if and only if both the encoder* $E_i$ *and generator* $\mathcal{G}_i$ *produce encodings such that the conditional expectation of the graph structure is identical to the marginal expectation:*

$$\mathbb{E}[\mathbf{g}|\tilde{\mathcal{E}}_{real}] = \mathbb{E}[\mathbf{g}] \quad and \quad \mathbb{E}[\mathbf{g}|\tilde{\mathcal{E}}_{syn}] = \mathbb{E}[\mathbf{g}].$$

---

[5]We note that $\mathcal{D}_i^k$ denotes the empirical label distribution of task j of client i, which can be stored as lightweight metadata (*e.g.*, label counts or label sets) without retaining raw data samples. This assumption is standard in replay-based federated continual learning (Abudukelimu et al., 2024; Qi et al., 2023).

[6]For readability and simplicity, since such a fundamental mathematical property of MSE loss is universal, we keep notations in a generic form, *i.e.*, $g$, $\tilde{\mathcal{E}}_{syn}$, $\tilde{\mathcal{E}}_{real}$, and etc.

*Proof.* We analyze the loss term for either component identically. Let $\tilde{\mathcal{E}}$ denote a noisy embedding from either source. From the Law of Total Variance, we have the decomposition:

$$\mathbb{V}[\mathbf{g}] = \mathbb{E}_{\tilde{\mathcal{E}}}[\mathbb{V}[\mathbf{g}|\tilde{\mathcal{E}}]] + \mathbb{V}_{\tilde{\mathcal{E}}}[\mathbb{E}[\mathbf{g}|\tilde{\mathcal{E}}]].$$

The term $\mathbb{V}[\mathbf{g}]$ represents the inherent variance of the graph dataset and is constant with respect to the generative models. To maximize the discriminator's error (the first term, $\mathbb{E}[\mathbb{V}[\mathbf{g}|\tilde{\mathcal{E}}]]$), the generative components must minimize the second term, $\mathbb{V}_{\tilde{\mathcal{E}}}[\mathbb{E}[\mathbf{g}|\tilde{\mathcal{E}}]]$.

Since variance is non-negative, the global minimum is 0, which is achieved if and only if the random variable $Z = \mathbb{E}[\mathbf{g}|\tilde{\mathcal{E}}]$ is a constant almost everywhere. Taking expectations over $\tilde{\mathcal{E}}$, this constant must equal the prior mean $\mathbb{E}[\mathbf{g}]$. Thus, at the global optimum, the embeddings contain no information about the specific graph instance $\mathbf{g}$, rendering them structurally invariant. □

This result implies that the proposed minimax game forces the system to align distributions based on shared structural properties. We summarize the implications for heterogeneity and forgetting in the following corollaries.

**Corollary A.6** (Mitigation of Statistical Heterogeneity). *At the global optimum,* $\tilde{\mathcal{E}}_{real} \perp\!\!\!\perp \mathbf{g}$, *i.e.,* $\mathbb{E}[\mathbf{g} \mid \tilde{\mathcal{E}}_{real}] = \mathbb{E}[\mathbf{g}]$. *This first-moment invariance implies that* $E_i$ *suppresses graph-induced bias at the level of conditional means, thereby reducing statistical heterogeneity across clients.*

**Corollary A.7** (Structural Consistency). *Since real and synthetic embeddings satisfy the same first-moment invariance condition, i.e.,* $\mathbb{E}[\mathbf{g} \mid \tilde{\mathcal{E}}] = \mathbb{E}[\mathbf{g}]$, $\mathcal{G}_i$ *achieves structural consistency with* $E_i$, *enabling generative replay to preserve latent-space structure and mitigate catastrophic forgetting.*

*Remark* A.8 (Robustness to DP Noise). Additive Laplace noise $\tau_\mathcal{E}$ adds constant variance but does not change the equilibrium. If the embedding is first-moment graph-invariant, *i.e.*, $\mathcal{E} \perp\!\!\!\perp \mathbf{g}$, then this invariance is preserved after adding independent DP noise.

*Remark* A.9 (Alignment of High-Order Moments). Theorem 4.2 shows that GFedCL achieves the first moment (mean) alignment. Higher-order moments (*e.g.*, mean and variance) are achievable via changing the discriminator's loss function (*e.g.*, from MSE to Gaussian NLL).

## A.3. Global Optimum of GFedCL

This subsection provides the equilibrium analysis for the complete GFedCL framework, involving the four-player game between the encoder $E_i$, generator $\mathcal{G}_i$, predictor $F_i$, and discriminator $D$. We require the system to achieve not just distributional alignment (structural invariance) but also semantic correctness (accurate label prediction).

**Theorem A.10.** *If the Encoder $E_i$, Generator $\mathcal{G}_i$, Predictor $F_i$, and Discriminator $D$ are trained to reach the global optimum, the system satisfies the following properties:*

1. ***Distribution Matching:** The expectations of marginal distributions of real and synthetic embeddings are identical: $\mathbb{E}[P(\mathcal{E}_{real})] = \mathbb{E}[P(\mathcal{E}_{syn})]$.*

2. ***Info Preservation:** The encoder $E_i$ retains all information about the label $y$ contained in the input $x$ and graph $\mathbf{g}$, i.e., $H(y|\mathcal{E}_{real}) = H(y|x, \mathbf{g})$.*

3. ***Synthetic Semantics:** The generator $\mathcal{G}_i$ produces features containing precise label information, i.e., $H(y|\mathcal{E}_{syn}) = 0$.*

*Proof.* **Part 1 (Distribution Matching):** This follows directly from Theorem A.5. The adversarial component of the loss forces the generator and encoder to align their output distributions to satisfy the structural invariance condition, resulting in $\mathbb{E}[P(\mathcal{E}_{\text{real}})] = \mathcal{E}[P(\mathcal{E}_{\text{syn}})]$.

**Part 2 & 3 (Information Preservation & Synthetic Semantics):** To prove these properties, we examine the predictor loss term $V_p$ defined in Eqn. (1) , which minimizes the Cross-Entropy (CE) loss. Recall that for a predictor output distribution $q(y|\mathcal{E})$ and true label distribution $p(y|\mathcal{E})$, the expected CE loss is related to Conditional Entropy $H(y|\mathcal{E})$ by:

$$\mathbb{E}[\mathcal{L}_{CE}] = H(y|\mathcal{E}) + D_{KL}(p(y|\mathcal{E})||q(y|\mathcal{E})). \quad (4)$$

Since the KL divergence is non-negative ($D_{KL} \geq 0$), the Cross-Entropy loss is strictly lower-bounded by the conditional entropy $H(y|\mathcal{E})$. Therefore, minimizing $V_p$ implies minimizing the uncertainty of $y$ given the embedding.

**For the Generator $\mathcal{G}_i(z, y, \mathbf{g})$:** The generator minimizes the synthetic prediction loss $V_{p,syn}^j$. Since the target label $y$ is provided explicitly as an input, there exists a mapping $\mathcal{G}_i$ such that $\mathcal{E}_{\text{syn}}$ perfectly determines $y$. At the global optimum, the predictor $F_i$ perfectly recovers this dependency, driving the loss to its theoretical minimum.

$$\min V_{p,syn} \implies H(y|\mathcal{E}_{\text{syn}}) \to 0.$$

**For the Encoder $E_i(x, \mathbf{g})$:** The encoder minimizes the real prediction loss $V_p^k$. Because $E_i$ processes $x$, the conditional entropy of the label given the embedding cannot be lower than the intrinsic uncertainty of the label given the raw data (Data Processing Inequality). Thus, the lower bound is:

$$H(y|\mathcal{E}_{\text{real}}) \geq H(y|x, \mathbf{g}).$$

However, since $E_i$ is trained to minimize $V_p^k$, it seeks to retain all discriminative features such that $H(y|\mathcal{E}_{\text{real}})$ approaches this lower bound. Furthermore, $E_i$ must simultaneously satisfy the alignment constraint from Part 1 ($\mathbb{E}[P(\mathcal{E}_{\text{real}})] = \mathcal{E}[P(\mathcal{E}_{\text{syn}})]$). Since $\mathcal{E}_{\text{syn}}$ has been proven to encode $y$ perfectly (entropy $\to 0$) in the synthetic domain, the aligned real embeddings must also separate classes to the extent allowed by the raw data. Therefore, at equilibrium, the encoder retains maximal semantic information:

$$H(y|\mathcal{E}_{\text{real}}) = H(y|x, \mathbf{g}).$$

$\square$

Theorem A.10 establishes that the optimal encoder retains all semantic information related to the label $y$ present in the data $x$, while simultaneously achieving structural alignment with the synthetic data distributions.

### A.4. Convergence Analysis of GFedCL

Compared to conventional FL methods that employ adversarial training (*i.e.*, generator versus discriminator), GFedCL preserves the same fundamental min-max training paradigm. The key modification is the privacy-preserving data transmission. Since the noise introduced by DP has a constant variance and zero mean, it acts as a regularization term but does not alter the convexity/concavity properties of the underlying loss landscape with respect to the first moments. Therefore, the convergence analysis established in previous work for federated adversarial learning (Rasouli et al., 2020) remains applicable to GFedCL.

## B. Additional Details and Experiments

### B.1. Definition of Evaluation Metrics

CL performance is commonly evaluated using two complementary metrics: *overall performance* (OP), which captures the average predictive quality across all tasks, and *backward transfer* (BWT), which quantifies the effect of learning new tasks on previously learned ones.

Overall performance reflects the model's global utility over the entire task sequence, while BWT isolates *forgetting*: a negative BWT indicates that learning later tasks degrades performance on earlier tasks, *i.e.*, *forgetting*.

In our setting, OP measures task-level performance (*e.g.*, prediction accuracy or mean-squared error), and BWT explicitly characterizes forgetting as the negative change in OP on past tasks.

**Overall Performance (OP).** We define OP as the average loss over all tasks and all clients:

$$\text{OP} := \frac{1}{K} \sum_{k=1}^{K} \frac{1}{N} \sum_{i=1}^{N} \mathcal{L}_i^k\left(\theta_i^k; D_i^k\right),$$

where $K$ denotes the total number of tasks, $N$ the number of clients, $\mathcal{L}_i^k$ the loss of client $i$ on task $k$, $\theta_i^k$ the model parameters after training up to task $k$, and $D_i^k$ the corresponding dataset.

**Backward Transfer (BWT).** Backward transfer measures the average performance change on previously learned tasks after training on subsequent ones. Formally, BWT is defined as

$$\text{BWT} := \frac{1}{K-1} \sum_{k=2}^{K} \frac{1}{k-1} \sum_{j=1}^{k-1} \left( \text{OP}_j^{(k)} - \text{OP}_j^{(j)} \right),$$

where $\text{OP}_j^{(k)}$ denotes the performance on task $j$ evaluated after completing training on task $k$, and $\text{OP}_j^{(j)}$ is the performance on task $j$ immediately after it is learned.

## B.2. Correlation between Relational Graphs and Data Distributions

To justify the design of the spatio-temporal graph, we examine whether the learned relational weights truly reflect the underlying data distributions. Intuitively, if two clients share similar data, they should be strongly connected in the graph, and if their distributions differ or drift apart over time, their edge weight should diminish. This subsection investigates that relationship empirically and demonstrates that GFedCL's graph captures both spatial heterogeneity and temporal evolution of data.

For each task $k$, we compute the pairwise *distribution distance* between clients. Depending on the data type, different metrics are employed: for image datasets (EMNIST-Letters, CIFAR-100, TinyImageNet), we estimate distributions in feature space using a trained feature extractor and compute the Jensen–Shannon divergence (JSD) or Maximum Mean Discrepancy (MMD) between the empirical distributions; for the US-States influenza dataset, where data are time-series of infection counts, we compute dynamic time warping (DTW) distances. Denote these distances by $d_{ij}^k$. We then compute the corresponding relational-graph weights $G_{ij}^k$ from the spatio-temporal mechanism.

**Correlation Analysis.** For each task, we calculate the Pearson correlation coefficient between $\{G_{ij}^k\}_{i \neq j}$ and $\{\exp(-d_{ij}^k)\}_{i \neq j}$ (so larger distance corresponds to smaller similarity). Across all datasets, we observe strong positive correlation, with mean correlation coefficients ranging from 0.41 to 0.66, indicating that larger graph weights correspond to smaller distributional differences. For example, on CIFAR-100, the average correlation is $0.52 \pm 0.08$, while on the US-States dataset, it is $0.66 \pm 0.04$. In contrast, using

only spatial attention (*i.e.*, ignoring temporal dynamics) reduces the correlation to about 0.45, and using random graphs yields correlations near zero. For reference and comprehensive understanding, Figure 6 presents the visualization of relational graphs and clients' data distributions from spatial and temporal aspects on *EMNIST-Letters*. Figure 7 also provides the trend of Pearson correlation across tasks.

**Temporal Dynamics.** To assess whether the temporal component captures concept drift, we examine how the correlation evolves over successive tasks. As tasks change (e.g., new influenza seasons or emerging COVID-19 variants), the underlying distributions shift and the graph weights adapt accordingly. We compute the *lagged correlation* between $G^k$ and $G^{k-1}$. When the distributions remain stable across tasks, the graph changes little, and the lagged correlation is high (above 0.9); when concept drift occurs, the correlation drops, reflecting that the graph correctly updates to reflect new relationships. This behavior is particularly evident in the US-States dataset, where seasonal influenza patterns vary year to year, and states that are connected in one season become less connected in the next as peaks occur at different times. Without temporal attention, the graph fails to track these shifts and replay uses outdated relationships, leading to inferior performance.

**Case Study.** Consider the EMNIST-Letters dataset, where clients correspond to different writers. Clients with similar handwriting styles or letter frequency distributions have high weights in $G^k$. When a client introduces new characters in later tasks, temporal attention becomes less similar to other temporal attention, lowering $G_{ij}^k$ and prompting GFedCL to allocate more synthetic data to align its distribution. Likewise, in the US-States dataset, clients representing geographically close states (e.g., New Jersey and Pennsylvania) receive high weights during the early flu season, but as outbreaks progress at different paces, the temporal attention component adjusts, redirecting influence to states whose infection curves are more aligned. These qualitative observations match the quantitative correlations described above.

**Implications for Model Performance.** The strong correlation between relational graphs and data distributions explains the improvements seen in GFedCL's ablation study. When the graph is removed or temporal attention is disabled, the replay mechanism cannot appropriately align distributions across clients and tasks, leading to larger errors and a greater performance drop. In contrast, by encoding both spatial and temporal similarities, GFedCL ensures that the synthetic data generation is conditioned on the most relevant sources, effectively mitigating catastrophic forgetting and statistical heterogeneity. These findings confirm that the spatio-temporal graph is not merely a heuristic but a

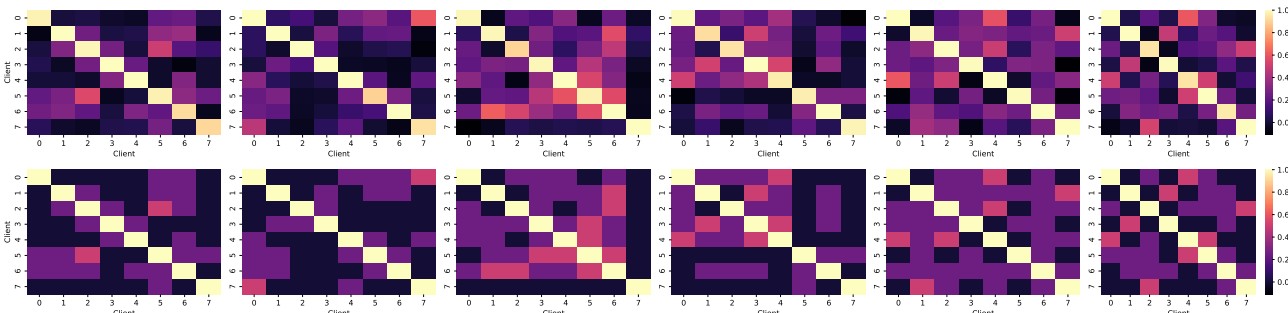

*Figure 6.* Visualization of relational graphs and clients' data distributions on EMNIST-Letters via heatmaps. The visualization presents the idea from spatial and temporal aspects simultaneously.

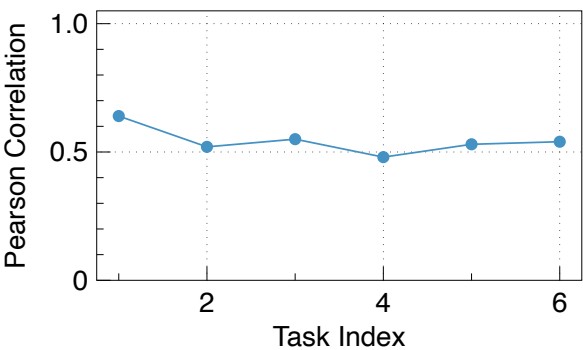

*Figure 7.* Pearson correlation between relational graphs and clients' data distributions. The results are taken over *EMNIST-Letters*. As shown in the figure, the high positive correlation suggests that relational graph does capture the spatial and temporal information desired.

principled representation of data relationships.

### B.3. Ablation Study

This subsection evaluates the impact of each component of GFedCL on model performance using the training datasets. The key components of GFedCL include relational graphs with spatial and temporal attention mechanisms, the replay mechanism, and the global discriminator. Relational graphs are first constructed using spatial attention and subsequently extended over time through temporal attention. As a result, it is not feasible to conduct an ablation study isolating spatial attention alone (*i.e.*, *w/o spatial attention*). Overall, the results support the effectiveness of incorporating additional information, such as spatial and temporal context, and of using a global discriminator in enhancing model performance in FCL, consistent with our theoretical analysis.

**With/without Temporal Attention:** Table 3 shows the effectiveness of involving temporal attention on distinguishing differences tasks.

**With/without Graph:** Table 3 shows the effectiveness of

relational graph on the replay and aligning task distributions. Implementing such a configuration involves removing the entire spatial and temporal attention mechanisms and local classifiers, and setting the discriminator's output to "real/fake."

**With/without Replay:** Table 3 shows the significance of the replay mechanism in mitigating catastrophic forgetting.

**With/without Global Discriminator:** Table 3 shows the effectiveness of global $D$ on aligning data distributions of tasks. Implementing such a configuration involves removing the global discriminator and deploying local discriminators.

**With/without DP:** Table 3 shows the impact of DP on model performance.

### B.4. Sensitivity Tests

For hyperparameter sensitivity:

- **Privacy Budget** ($\epsilon \in \{0.1, 0.5, 1\}$)**:** Table 4 shows the model performance of GFedCL under different privacy budgets, ranging from strong privacy preservation ($\epsilon = 0.1$) to common privacy preservation ($\epsilon = 1$).

- **Balance Term** ($\lambda \in \{0.5, 0.7, 1\}$)**:** Table 4 shows the model performance of GFedCL under different balance terms.

- **Epoch** (#**epoch** $\in \{1, 5, 10\}$**:** Table 4 shows the model performance of GFedCL under different training epochs.

### B.5. Quality of Synthetic Samples

To assess the fidelity and diversity of the synthetic data, we employ two commonly used metrics in generative modeling, FID (Heusel et al., 2017) and IS (Salimans et al., 2016). Figure 8(a) presents the IS results, while Figure 8(b) shows the FID scores.

*Table 3.* Ablation study of GFedCL on synthetic and real-world datasets. "GFedCL" denotes the original model, "w/o DP" removes the DP, and "w/o index" replaces the continuous domain index with the categorical domain index (*e.g.*, "0 → source" and "1 → target").

| Method | EMNIST-Letters Accuracy (%)↑ | CIFAR-100 Accuracy (%)↑ | TinyImageNet Accuracy (%)↑ | TCGA-BRCA Accuracy (%)↑ | US-States MSE↓ |
|---|---|---|---|---|---|
| GFedCL | $77.23 \pm 2.76$ | $73.67 \pm 3.42$ | $77.39 \pm 3.02$ | $75.32 \pm 2.19$ | $2.41 \pm 0.18$ |
| w/o DP | $78.45 \pm 1.98$ | $75.09 \pm 2.74$ | $79.67 \pm 2.54$ | $77.04 \pm 2.24$ | $2.28 \pm 0.14$ |
| w/o Graph | $53.45 \pm 2.88$ | $50.20 \pm 3.21$ | $53.12 \pm 2.89$ | $50.98 \pm 2.55$ | $3.55 \pm 0.31$ |
| w/o Temporal Attention | $60.17 \pm 3.54$ | $58.40 \pm 3.33$ | $61.40 \pm 2.54$ | $58.06 \pm 3.87$ | $2.57 \pm 0.61$ |
| w/o Replay | $42.90 \pm 4.29$ | $39.70 \pm 3.10$ | $42.95 \pm 1.94$ | $41.50 \pm 3.01$ | $3.75 \pm 0.42$ |
| w/o Global $\mathcal{G}$ | $65.10 \pm 2.29$ | $62.70 \pm 2.10$ | $65.95 \pm 1.94$ | $63.50 \pm 2.01$ | $2.95 \pm 0.21$ |

*Table 4.* Sensitivity analysis of FedIndex with respect to privacy budget $\epsilon$, regularization weight $\lambda$, and training epochs. We report the mean $\pm$ standard deviation over multiple runs. Accuracy is used for classification tasks, and MSE is reported for US-States.

| Setting | Value | EMNIST-Letters Accuracy (%)↑ | CIFAR-100 Accuracy (%)↑ | TinyImageNet Accuracy (%)↑ | TCGA-BRCA Accuracy (%)↑ | US-States MSE ↓ |
|---|---|---|---|---|---|---|
| $\epsilon$ | 0.15 | $76.09 \pm 2.66$ | $73.11 \pm 3.91$ | $73.10 \pm 3.95$ | $73.96 \pm 2.42$ | $2.33 \pm 0.19$ |
| | 0.30 | $76.70 \pm 2.99$ | $73.43 \pm 5.62$ | $74.46 \pm 3.42$ | $75.87 \pm 2.08$ | $2.32 \pm 0.18$ |
| | 0.50 | $77.05 \pm 2.90$ | $73.57 \pm 4.89$ | $75.57 \pm 3.67$ | $77.49 \pm 2.25$ | $2.37 \pm 0.21$ |
| | 1.00 | $77.23 \pm 2.76$ | $73.67 \pm 3.42$ | $77.39 \pm 3.02$ | $75.32 \pm 2.19$ | $2.41 \pm 0.18$ |
| $\lambda$ | 0.3 | $76.97 \pm 2.10$ | $75.56 \pm 3.42$ | $75.18 \pm 2.92$ | $78.98 \pm 2.50$ | $2.45 \pm 0.16$ |
| | 0.5 | $78.11 \pm 2.64$ | $73.17 \pm 5.62$ | $73.89 \pm 3.07$ | $75.93 \pm 2.03$ | $2.33 \pm 0.21$ |
| | 0.7 | $78.81 \pm 2.21$ | $74.67 \pm 3.18$ | $78.83 \pm 3.79$ | $77.65 \pm 2.28$ | $2.39 \pm 0.15$ |
| | 1.0 | $77.23 \pm 2.76$ | $73.67 \pm 3.42$ | $77.39 \pm 3.02$ | $75.32 \pm 2.19$ | $2.41 \pm 0.18$ |
| Epoch | 1 | $77.23 \pm 2.76$ | $73.67 \pm 3.42$ | $77.39 \pm 3.02$ | $75.32 \pm 2.19$ | $2.41 \pm 0.18$ |
| | 5 | $76.20 \pm 2.63$ | $73.27 \pm 7.33$ | $76.89 \pm 2.34$ | $75.94 \pm 1.98$ | $2.33 \pm 0.13$ |
| | 10 | $76.24 \pm 3.17$ | $73.83 \pm 4.89$ | $77.06 \pm 3.40$ | $78.07 \pm 0.90$ | $2.42 \pm 0.19$ |

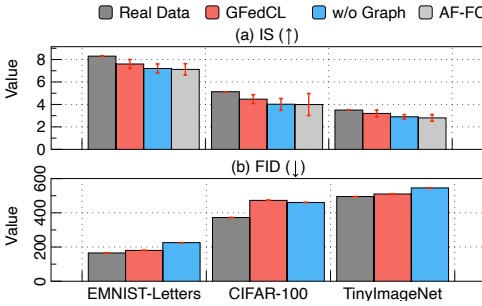

*Figure 8.* IS and FID scores of GFedCL, GFedCL w/o graph, and AF-FCL on training datasets.

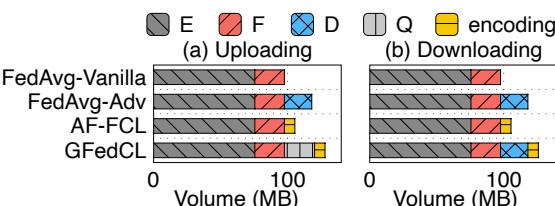

*Figure 9.* Communication overhead of GFedCL.

### B.6. Communication Overhead

This subsection presents the communication overhead (in bytes) of GFedCL per communication round, in comparison to conventional FL methods and state-of-the-art approaches such as AF-FCL. Specifically, we consider two variants of

FedAvg as baselines: (1) FedAvg-Vanilla: The standard FedAvg framework, comprising a feature extractor (encoder) and a classifier. (2) FedAvg-Adv: A variant of FedAvg augmented with a local discriminator for adversarial training. To ensure fairness, component sizes are kept identical across methods when the components serve the same functional role, namely, "E" for encoder, "F" for predictor, "D" for discriminator, and "Q" for classifier. As shown in Figure 9, GFedCL introduces only a modest increase in communication overhead compared to FedAvg-Adv, primarily due

to the transmission of pre-trained classifiers and encodings. The additional information required for graph construction consists of the classifier updates from each client, which amount to roughly 20 MB for ResNet-18. By comparison, the standard transmission of full model updates is about 100 MB. Thus, GFedCL adds only around 20% extra storage overhead on the server for graph construction, which is unlikely to be a bottleneck for modern server systems.

### B.7. Computational Overhead

This subsection outlines the computational overhead (in seconds) of GFedCL in comparison to conventional FL methods and other baselines. The additional overhead of GFedCL primarily arises from graph generation and discriminator training on the server side. As with previous analyses, we include two variants of conventional FL methods: FedAvg-Vanilla and FedAvg-Adv. Table 5 reports the average training time per communication round for GFedCL and the baseline methods.

The computational overhead results in Table 5 demonstrate that graph construction introduces only a 20–25% increase in computation time (GFedCL vs. FedAvg-Adv). For instance, on the US-States dataset, FedAvg takes 4.88s per round, while GFedCL takes 5.78s; the additional 0.9s ( 20%) is strictly for server-side graph construction. We believe this cost remains highly affordable under current server configurations for two key reasons: (1) the computation is performed centrally on the server, sparing resource-constrained clients, and (2) the operations process only compact, aggregated model information, which is easily parallelized (e.g., sparse matrix multiplications) on modern hardware. Therefore, although graph construction does incur an extra cost, we do not expect it to become a severe bottleneck in realistic deployments.

### B.8. KS Tests for the Effectiveness of DP

Prior work (Wang et al., 2018; Awan & Wang, 2023) has analyzed the use of Kolmogorov–Smirnov (KS) tests for evaluating the effectiveness of DP mechanisms. Following this line of work, we apply two-sample KS tests with privacy budgets $\epsilon \in \{0.1, 0.5, 1\}$ to assess the extent to which DP noise obscures sensitive information in GFedCL. Specifically, we compare the distributions of clean embeddings and their DP-noised counterparts observed by the server or curious clients.

Table 6 reports the KS statistics and corresponding p-values for embeddings and domain indices under different privacy levels. A low p-value ($\leq 0.05$) indicates that the null hypothesis of identical distributions is rejected, meaning that the noised and clean representations are statistically distinguishable. Conversely, a high p-value suggests insufficient evidence to distinguish the two distributions, implying that

DP noise effectively masks the underlying data and limits the feasibility of DIA or MIA. Across our experiments, smaller privacy budgets (*e.g.*, $\epsilon \leq 1$) consistently yield higher p-values, indicating stronger privacy protection. [7]

### B.9. Group Division of *US-States*

Health and Human Services (HHS), one department within the Centers for Disease Control (CDC), divides U.S. states into regional groups based on geographic proximity (U.S. Department of Health and Human Services, Office of Intergovernmental and External Affairs, 2025). Inspired by this administrative structure and real-world ILI distributions (Centers for Disease Control and Prevention, 2025), we adapt the division to construct the *US-States* scenario, where seven clients represent seven groups of states clustered by geographic connectivity. Specifically, Table 7 and Figure 10 show the group details and visualization, respectively.

### B.10. Trade-off between Replay Size and Model Performance

Previous studies often generate an equal amount of synthetic data as the real data during replay. However, this strategy incurs significant computational and communication overhead, which may not be feasible for edge devices with limited resources (*e.g.*, smartwatches and internet of things (IoT) devices). Therefore, it is important to explore the trade-off between the quantity of generated synthetic data and overall model performance, including efficiency and computational overhead. Such an analysis can help users strike a practical balance between performance and resource consumption. Table 8 presents the results under three replay configurations: 0%, 50%, and 100% of the real data size.

### B.11. Selection of Classifier $Q$

GFedCL adopts ResNet18 (He et al., 2016) as the classifier $Q$ due to its strong compatibility with image classification tasks. However, the graph generation process in GFedCL is model-agnostic. Prior work (Sattler et al., 2021) has shown that, given sufficient local training data and communication rounds, model updates can effectively reflect the underlying data distributions even when using a simple CNN. To empirically validate that this insight also holds for GFedCL, as

---

[7]The dataset sensitivity in GFedCL is set to 1, which is a common choice in DP analysis. However, in real-world scenarios, the dataset sensitivity may vary depending on the adopted threat model, *e.g.*, utilizing attacks other than MIA and DIA. Therefore, the resulting privacy budget may not strictly match the one analyzed in GFedCL. Nevertheless, changing the dataset sensitivity only rescales the effective privacy budget and does not affect the equilibrium analysis or other theoretical conclusions of GFedCL.

*Table 5.* Computational overhead (s) of GFedCL on training datasets. Each table element refers to the average training time per communication round of each method.

| Model | EMNIST-Letters | CIFAR-100 | TinyImageNet | TCGA-BRCA | US-States |
|---|---|---|---|---|---|
| GLFC | 9.33 | 14.61 | 31.25 | 11.45 | 4.94 |
| FedAvg-Adv | 7.33 | 12.11 | 27.25 | 9.89 | 4.88 |
| FedCIL | 9.67 | 14.61 | 33.25 | 10.07 | 5.88 |
| MFCL | 12.17 | 20.53 | 40.70 | 15.77 | 5.10 |
| AF-FCL | 9.17 | 16.22 | 35.14 | 13.25 | 6.12 |
| GFedCL | 11.41 | 17.34 | 36.11 | 12.37 | 5.78 |

*Table 6.* Kolmogorov–Smirnov (KS) test results under different privacy budgets $\epsilon$ (smaller $\epsilon$ indicates stronger privacy). We report KS statistics (larger is better) here. All corresponding $p$-values are less than 0.05.

| $\epsilon$ | Method | EMNIST-Letters | CIFAR-100 | TinyImageNet | TCGA-BRCA | US-States |
|---|---|---|---|---|---|---|
| $\epsilon = 1$ | Encoding | 0.32 | 0.30 | 0.29 | 0.38 | 0.35 |
| | Graph | 0.28 | 0.26 | 0.25 | 0.33 | 0.31 |
| $\epsilon = 0.5$ | Encoding | 0.34 | 0.32 | 0.31 | 0.40 | 0.37 |
| | Graph | 0.29 | 0.27 | 0.26 | 0.34 | 0.32 |
| $\epsilon = 0.3$ | Encoding | 0.36 | 0.34 | 0.33 | 0.42 | 0.39 |
| | Graph | 0.30 | 0.28 | 0.27 | 0.35 | 0.33 |
| $\epsilon = 0.15$ | Encoding | 0.38 | 0.36 | 0.35 | 0.44 | 0.41 |
| | Graph | 0.31 | 0.29 | 0.28 | 0.36 | 0.34 |
| $\epsilon = 0.1$ | Encoding | 0.40 | 0.38 | 0.36 | 0.45 | 0.42 |
| | Graph | 0.32 | 0.30 | 0.29 | 0.37 | 0.35 |
| $\epsilon = 0.05$ | Encoding | 0.42 | 0.40 | 0.38 | 0.47 | 0.44 |
| | Index | 0.33 | 0.31 | 0.30 | 0.38 | 0.36 |
| $\epsilon = 0.01$ | Encoding | 0.45 | 0.43 | 0.41 | 0.50 | 0.47 |
| | Index | 0.35 | 0.33 | 0.32 | 0.40 | 0.38 |

is shown in Table 9 we replace the default classifier $Q$ with alternative architectures and evaluate the resulting model performance.

## C. Discussions

This section discusses several concerns and practical issues when designing GFedCL.

### C.1. Details of Spatio-Temporal Attention Mechanism

A key novelty in **GFedCL** is the *spatio-temporal relational graph* used to guide replay. Unlike prior FL–CL methods that rely solely on class labels or simple heuristics, GFedCL explicitly models how clients influence one another both spatially (across clients) and temporally (across tasks). This graph is constructed on the server and then broadcasts to clients before each round. To avoid the feedback loops that

can occur in adversarial settings, graph construction is *decoupled* from training: the server extracts information from uploaded classifier updates $w_i^k$ for the current task, builds the graph, and sends only the appropriate row $g_i^k$ back to client $i$. Once the graph is built, the local encoder–generator–predictor triplets $(E_i, G_i, F_i)$ use it only as conditioning information and do not influence the graph during that round. This design permits the graph to capture client interactions without being polluted by synthetic samples.

Let $C_i$ and $C_j$ be two clients during task $k$. Each client trains a classifier $Q$ on its local data and uploads the model update $w_i^k$ to the server. To quantify spatial similarity, the server computes an attention score $\alpha_{ij}^k$ for each pair of clients using a learnable function $a(\cdot)$. In practice, $a(\cdot)$ can be implemented as a graph-attention layer (GAT):

$$\alpha_{ij}^k = \text{softmax}_j\big(a(w_i^k, w_j^k)\big), \qquad \alpha_{ij}^k \in (0, 1).$$

*Table 7.* Forty-nine U.S. states are clustered into seven groups based on geographic connectivity. Hawaii, which is typically not contiguous with other states, is included in Group 7 but omitted from the visualization for clarity.

| Group 1 | Group 2 | Group 3 | Group 4 | Group 5 | Group 6 | Group 7 |
| --- | --- | --- | --- | --- | --- | --- |
| Maine | New Jersey | North Carolina | Michigan | Louisiana | Montana | California |
| New Hampshire | Delaware | South Carolina | Indiana | Mississippi | Wyoming | Oregon |
| Vermont | Maryland | Georgia | Illinois | Nebraska | Colorado | Washington |
| Massachusetts | Pennsylvania | Florida | Wisconsin | Kansas | New Mexico | Arizona |
| Rhode Island | Virginia | Alabama | Minnesota | Oklahoma | Utah | Idaho |
| Connecticut | West Virginia | Tennessee | Iowa | Texas | North Dakota | Nevada |
| New York | Ohio | Kentucky | Missouri | Arkansas | South Dakota | Hawaii |

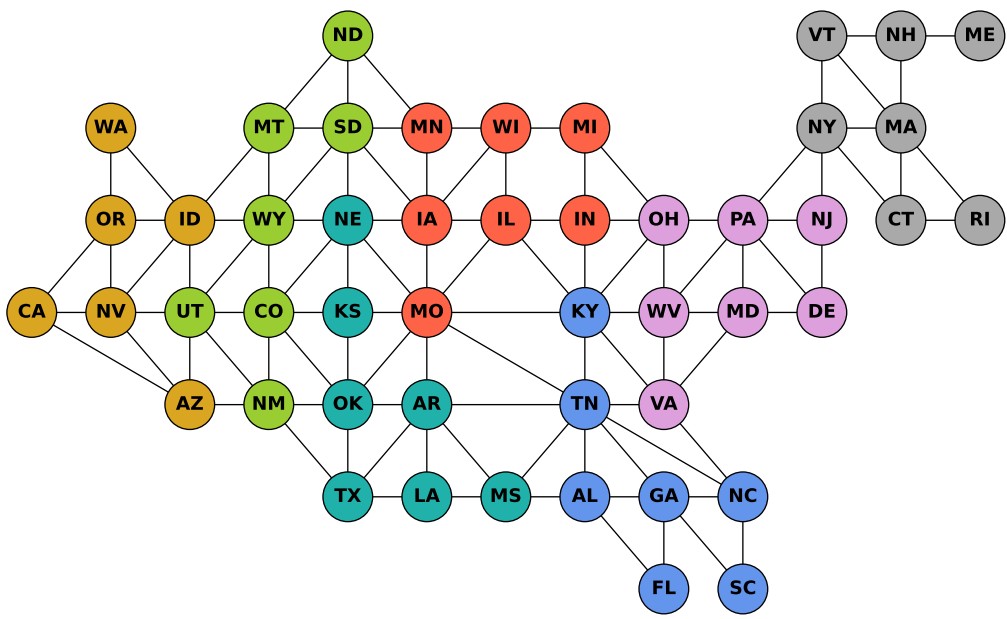

*Figure 10.* Visualization of the division of forty-nine states into seven groups. The grouping is based on geographic connectivity, with the exception of Hawaii, which is typically disconnected from the other forty-eight states. Although Hawaii is included in Group 7, it is excluded from the visualization.

Here, $a(w_i^k, w_j^k)$ computes a compatibility score between the two updates—for example by concatenating features, projecting them through a shared weight matrix, applying a non-linearity and a learned weight vector (Veličković *et al.*, 2018). The softmax ensures the scores across all neighbours sum to one, yielding a *spatially normalized* attention vector. Intuitively, $\alpha_{ij}^k$ will be large when clients $i$ and $j$ exhibit similar classifier updates, indicating that their data distributions in the current task are close; it will be small when the updates differ markedly.

More concretely, we implement the compatibility function $a(\cdot)$ following the GAT formulation. Each model update $w_i^k \in \mathbb{R}^d$ is projected into a shared latent space via a linear transformation $W \in \mathbb{R}^{d' \times d}$. The transformed vectors $[Ww_i^k \| Ww_j^k]$ are concatenated and passed through a learn-

able weight vector $v_a \in \mathbb{R}^{2d'}$, yielding a normalized score

$$e_{ij}^k = \sigma\big(v_a^\top [Ww_i^k \| Ww_j^k]\big),$$

where $\sigma(\cdot)$ is a non-linear activation such as LeakyReLU. The spatial attention coefficient is then obtained by normalizing across all neighbors:

$$\alpha_{ij}^k = \frac{\exp(e_{ij}^k)}{\sum_{j' \in [N]} \exp(e_{ij'}^k)}.$$

This formulation is fully differentiable and allows the network to learn which neighbors are most relevant during each task.

Spatial proximity alone cannot capture concept drift across tasks, so GFedCL augments the graph with a *temporal* attention score. Let $H_i^k$ be a sliding-window history

*Table 8.* Trade-off of synthetic data size versus model performance and computational overhead across three replay configurations: 0%, 50%, and 100%. We report the mean $\pm$ standard deviation for Accuracy and MSE. *Acc.* denotes Accuracy.

| Ratio | Model | EMNIST-Letters | | CIFAR-100 | | TinyImageNet | | TCGA-BRCA | | US-States | |
|---|---|---|---|---|---|---|---|---|---|---|---|
| | | Acc. (%) ↑ | Time (s) ↓ | Acc. (%) ↑ | Time (s) ↓ | Acc. (%) ↑ | Time (s) ↓ | Acc. (%) ↑ | Time (s) ↓ | MSE (×10⁻²) ↓ | Time (s) ↓ |
| 100% | GLFC | $43.47 \pm 2.14$ | 9.33 | $46.08 \pm 2.70$ | 14.61 | $42.93 \pm 2.88$ | 31.25 | $59.74 \pm 1.97$ | 11.45 | $3.40 \pm 0.22$ | 4.94 |
| | FedAvg-Adv | $25.67 \pm 4.49$ | 7.33 | $32.05 \pm 4.51$ | 12.11 | $25.41 \pm 6.55$ | 27.25 | $32.70 \pm 5.20$ | 9.89 | $3.95 \pm 0.20$ | 4.88 |
| | FedCIL | $38.95 \pm 4.53$ | 9.67 | $39.47 \pm 3.24$ | 14.61 | $45.75 \pm 5.26$ | 33.25 | $56.59 \pm 4.75$ | 10.07 | $3.14 \pm 0.31$ | 5.88 |
| | MFCL | $52.66 \pm 3.24$ | 12.17 | $37.09 \pm 3.76$ | 20.53 | $47.82 \pm 3.09$ | 40.70 | $54.81 \pm 2.89$ | 15.77 | $2.74 \pm 0.24$ | 5.10 |
| | AF-FCL | $41.22 \pm 4.51$ | 9.17 | $43.27 \pm 4.77$ | 16.22 | $54.91 \pm 3.72$ | 35.14 | $67.44 \pm 2.59$ | 13.25 | $3.84 \pm 0.19$ | 6.12 |
| | **GFedCL** | $\mathbf{77.23 \pm 2.76}$ | 11.41 | $\mathbf{73.67 \pm 3.42}$ | 17.34 | $\mathbf{77.39 \pm 3.02}$ | 36.11 | $\mathbf{75.32 \pm 2.19}$ | 12.37 | $\mathbf{2.41 \pm 0.18}$ | 5.78 |
| 50% | GLFC | $37.19 \pm 2.51$ | 8.11 | $38.90 \pm 3.01$ | 12.93 | $36.60 \pm 3.15$ | 28.45 | $54.33 \pm 2.25$ | 9.94 | $3.67 \pm 0.28$ | 4.41 |
| | FedAvg-Adv | $22.13 \pm 4.88$ | 6.52 | $28.34 \pm 4.90$ | 10.94 | $21.78 \pm 6.90$ | 23.66 | $29.80 \pm 5.40$ | 8.60 | $4.24 \pm 0.25$ | 4.36 |
| | FedCIL | $33.40 \pm 4.95$ | 8.35 | $33.20 \pm 3.62$ | 12.73 | $40.84 \pm 5.70$ | 30.10 | $52.01 \pm 5.01$ | 8.76 | $3.45 \pm 0.35$ | 5.23 |
| | MFCL | $46.38 \pm 3.66$ | 10.28 | $32.33 \pm 4.10$ | 17.92 | $42.18 \pm 3.44$ | 36.98 | $49.66 \pm 3.20$ | 13.72 | $2.98 \pm 0.28$ | 4.79 |
| | AF-FCL | $37.35 \pm 4.90$ | 8.43 | $37.38 \pm 5.15$ | 14.35 | $49.68 \pm 4.11$ | 32.91 | $61.20 \pm 2.90$ | 11.53 | $4.10 \pm 0.25$ | 5.48 |
| | **GFedCL** | $\mathbf{69.27 \pm 2.85}$ | 10.02 | $\mathbf{66.34 \pm 3.61}$ | 15.86 | $\mathbf{71.16 \pm 3.20}$ | 33.22 | $\mathbf{69.80 \pm 2.45}$ | 10.76 | $\mathbf{2.66 \pm 0.21}$ | 5.13 |
| 0% | GLFC | $29.84 \pm 2.80$ | 6.89 | $30.85 \pm 3.44$ | 11.03 | $28.56 \pm 3.50$ | 24.33 | $43.12 \pm 2.60$ | 8.46 | $4.07 \pm 0.35$ | 3.87 |
| | FedAvg-Adv | $17.97 \pm 5.20$ | 5.89 | $22.52 \pm 5.33$ | 9.82 | $17.61 \pm 7.40$ | 20.51 | $20.73 \pm 5.80$ | 7.32 | $4.74 \pm 0.33$ | 3.91 |
| | FedCIL | $26.91 \pm 5.30$ | 7.34 | $26.24 \pm 4.11$ | 11.12 | $34.54 \pm 6.10$ | 26.45 | $41.22 \pm 5.40$ | 7.45 | $3.89 \pm 0.40$ | 4.87 |
| | MFCL | $40.16 \pm 4.01$ | 9.46 | $26.90 \pm 4.55$ | 15.53 | $35.43 \pm 3.88$ | 31.90 | $40.01 \pm 3.55$ | 11.67 | $3.24 \pm 0.32$ | 4.34 |
| | AF-FCL | $33.16 \pm 5.25$ | 7.10 | $32.84 \pm 5.60$ | 12.18 | $44.52 \pm 4.45$ | 30.25 | $54.33 \pm 3.22$ | 9.80 | $4.41 \pm 0.30$ | 4.85 |
| | **GFedCL** | $\mathbf{59.82 \pm 3.12}$ | 8.94 | $\mathbf{41.46 \pm 3.98}$ | 14.22 | $\mathbf{63.42 \pm 3.44}$ | 31.87 | $\mathbf{61.50 \pm 2.80}$ | 9.15 | $\mathbf{2.43 \pm 0.22}$ | 4.49 |

*Table 9.* Model efficiency of GFedCL using different classifiers across various datasets. *Acc.* denotes Accuracy.

| Classifier | EMNIST-Letters | | CIFAR-100 | | TinyImageNet | | TCGA-BRCA | | US-States |
|---|---|---|---|---|---|---|---|---|---|
| | Acc. (%) ↑ | Forgetting (%) ↓ | Acc. (%) ↑ | Forgetting (%) ↓ | Acc. (%) ↑ | Forgetting (%) ↓ | Acc. (%) ↑ | Forgetting (%) ↓ | MSE (×10⁻²) ↓ |
| ResNet18 | 47.44 | 5.34 | 71.46 | 4.78 | 79.63 | 6.34 | 79.63 | 6.34 | 2.44 |
| ResNet50 | 49.16 | 5.14 | 73.21 | 3.98 | 82.15 | 7.88 | 79.63 | 6.34 | 2.42 |
| ResNet101 | 50.71 | 4.79 | 73.68 | 4.28 | 83.47 | 5.78 | 79.63 | 6.34 | 2.43 |

of client $i$'s previous relational-graph rows, i.e., $H_i^k = [\mathbf{g}_i^k, \mathbf{g}_i^{k-1}, \ldots, \mathbf{g}_i^{k-m}]$ for window length $m$. The server computes a temporal alignment score between clients $i$ and $j$ as

$$\beta_{ij}^k = \mathrm{softmax}_j\big(b(H_i^k, H_j^k, m)\big), \quad \beta_{ij}^k \in (0, 1),$$

where $b(\cdot)$ is an attention function. The sliding window ensures that the graph reflects *recent* dynamics, such as viral mutations or seasonality, without being dominated by distant tasks. Larger $\beta_{ij}^k$ indicates that the temporal evolution of client $j$'s distribution closely mirrors that of client $i$.

We instantiate $b(\cdot)$ as a scaled dot-product attention over the history vectors, analogous to the temporal attention proposed by Zhuang *et al.* (2021). Each history vector $H_i^k \in \mathbb{R}^{(m+1)N}$ is projected into query and key representations:

$$q_i^k = W_Q H_i^k, \quad k_j^k = W_K H_j^k,$$

where $W_Q$ and $W_K$ are learnable matrices. The normalized temporal score between clients $i$ and $j$ is

$$f_{ij}^k = \frac{\langle q_i^k, k_j^k \rangle}{\sqrt{d_k}},$$

where $d_k$ is the dimensionality of the key vectors. The temporal attention weights are obtained via a softmax:

$$\beta_{ij}^k = \frac{\exp(f_{ij}^k)}{\sum_{j' \in [N]} \exp(f_{ij'}^k)}.$$

The final relational graph for task $k$ is obtained by element-wise multiplication of the spatial and temporal scores:

$$G_{ij}^k = \alpha_{ij}^k \beta_{ij}^k \in (0, 1).$$

Because both $\alpha_{ij}^k$ and $\beta_{ij}^k$ are normalized, $G^k$ encodes how strongly client $j$ should influence client $i$'s replay. Multiplying the two scores ensures that large edges occur only when clients are similar *both* in their current task and in their historical patterns. The modular nature of this design allows extensions, such as incorporating graph-theoretic metrics, such as betweenness centrality, into $a(\cdot)$, or using alternative temporal kernels that emphasize recent tasks more strongly.

In practice, graph construction proceeds as follows: (1) each client trains classifier $Q$ locally and uploads $w_i^k$ to the server; (2) the server computes $\alpha^k$ from the current $w_i^k$ across all clients; (3) it updates the sliding-window histories $H_i^k$ and computes $\beta^k$; (4) it forms $G^k = \alpha^k \odot \beta^k$ and sends each row $g_i^k$ back to client $i$. Because only $N^2$ scores are computed per round and the window $m$ is modest (e.g., $m \in \{3, 5\}$), graph construction incurs little overhead relative to local training. Decoupling graph construction from the adversarial training loop avoids reinforcing spurious correlations and makes GFedCL robust to feedback loops.

## C.2. Feasibility of Uploading Features to Server for Training

Existing studies on FL (Peng et al., 2025) and federated split learning (FSL) (Thapa et al., 2022) are faced with similar challenges in terms of extra data transmission. Thus, to enable FL for specific objectives, such as maximizing the discriminator's distinguishing ability, it is common and acceptable to transmit features to the server for training. Nevertheless, previous study in FCL also employs a global normalizing flow model on the server to process latent space representations extracted by local clients, proving the feasibility of uploading features to the server for training (Abudukelimu et al., 2024).

## C.3. Rationale Behind Generating Relational Graphs Instead of Leveraging Explicit Graphs

Explicit graphs (*e.g.*, geometric graphs of states) typically encode a specific type of relational information, such as spatial connectivity, and are thus limited in representational scope for particular scenarios. While useful in narrow contexts, such graphs are insufficient for capturing the nuanced spatial and temporal relationships required in GFedCL. Therefore, instead of relying on predefined explicit graphs, we adopt a spatial- and temporal-aware mechanism to quantify each client's influence on others' downstream tasks, allowing the resulting relational graph to more effectively reflect dynamic inter-client relationships over time.

Similarly, although prior work (Sattler et al., 2021) demonstrates that cosine similarity between client model updates can reflect similarities in their data distributions, the resulting similarity matrix offers only a partial view and lacks the expressiveness needed to fully model the complex interdependencies among clients.

## C.4. Impact of FL Training Dynamics on Theoretical Analysis

Although GFedCL adopts a similar analytical framework as prior studies (Donahue et al., 2017; Zhao et al., 2019), namely, analyzing component behavior at optimality, those studies are conducted under centralized settings, whereas GFedCL operates in a FL environment. As such, our analysis must account for the unique training dynamics introduced by FL, particularly the effects of model aggregation.

To this end, we develop our analytical strategy through the following steps: (1) Identify the key property that GFedCL should exhibit at optimality, specifically, that synthetic data encodings become indistinguishable from real data encodings; (2) Design a learning framework capable of achieving this property, namely through an adversarial game between local encoders and discriminators; (3) Examine how FL training dynamics affect this framework. In particular, aggregating local discriminators across clients prevents the resulting global discriminator from achieving the same distinguishing capability as a centrally initialized discriminator, due to factors such as non-IID data and local update drift; (4) Adapt the learning framework to mitigate these effects by deploying a global discriminator on the server from the outset, rather than relying on locally trained discriminators.

Following this reasoning, we construct an adversarial learning framework consisting of a client-specific encoder–generator-predictor tuple and a shared global discriminator, which is robust to the aggregation-induced challenges inherent in FL.

## C.5. Variants of Relational Graphs

Beyond the current design based on spatial and temporal attention mechanisms, the relational graph generation can be extended with alternative scoring strategies. For example, spatial attention could incorporate graph metrics such as betweenness centrality (Freeman, 1977) to reflect pairwise node distances and structural importance.

## C.6. Strategies to Mitigate Feedback Loop and Model Collapse

Previous studies (Lucic et al., 2018; van de Ven & Tolias, 2018) have identified the issue of *feedback loops* in generative adversarial learning, which can lead to model collapse. A feedback loop arises from the iterative training dynamic in adversarial setups, where synthetic data is generated for training, and the resulting trained components are then used to generate more synthetic data. This cyclic dependency can amplify errors over time, ultimately resulting in model collapse. To address this, numerous methods have been proposed to stabilize training and mitigate collapse (Gulrajani et al., 2017).

In contrast to conventional adversarial learning, GFedCL involves an additional generation process, the generation of relational graphs, which could similarly introduce a *feedback loops* if not handled carefully. To prevent this, we explicitly decouple relational graph generation from the synthetic data generation process. Specifically, relational graphs in GFedCL are derived solely from real data, thereby avoiding the introduction of additional *feedback loops* and reducing the risk of model collapse.

## C.7. Incorporation of GFedCL With Security Methods Other Than DP

GFedCL raises additional privacy concerns compared to conventional FL methods, such as FedAvg, by transmitting data encodings between clients and the server. In addition to model inversion attacks such as DIA and MIA, such transmission of data encodings may also be vulnerable to model

poisoning attacks where attackers would transmit poisoned data encodings to influence training and thus degrade overall performance. To defend against model poisoning attacks, GFedCL can incorporate safeguards such as Krum (Blanchard et al., 2017) to select the most typical data encodings and then identify malicious poisoned data encodings. Specifically, as the encoders extract the domain-invariant features, the distances between benign data encodings are typically smaller than those between poisoned data encodings. Hence, incorporating Krum could help GFedCL defend against model poisoning attacks that aim at degrading overall model performance (*e.g.*, classification accuracy).

### C.8. Deployment challenges

Previous work (Li et al., 2020a) has highlighted several deployment challenges faced by federated learning (FL) in real-world scenarios. In response, numerous studies (Li et al., 2020b) have proposed methods to mitigate these challenges. In this work, we focus on the unique deployment challenges introduced by GFedCL, which are not typically encountered in conventional FL methods. Specifically, the generation of relational graphs and the additional data transmission involved in GFedCL introduce increased computational and communication overhead, which may exacerbate resource constraints during real-world deployment.

To address this issue, we first empirically quantify the computational and communication overhead of GFedCL in comparison to conventional FL methods such as FedAvg. Furthermore, we propose a practical approach to mitigate this overhead using data compression techniques, principal component analysis (PCA). This is motivated by the observation that the latent space representations used in GFedCL are typically sparse, with most elements being close to or even equal to zero. As a result, applying PCA allows for dimensionality reduction while preserving critical information, thereby reducing overhead without significantly compromising model performance.

## D. Pseudo Code

This section provides the pseudo-code of Algorithms 1 and 2, describing the workflow of GFedCL and the DP mechanism

## E. Configuration

### E.1. FCL Setup

Table 10 provides details of FCL setup on training datasets.

### E.2. Hyperparameter

Table 11 provides details of hyperparameter selection on training datasets.

## F. Reproduction

The code is available at `https://github.com/IntelliSys-Lab/GFedCL`.

---

**Algorithm 1** GFedCL

---

**Input:** initial encoder, generator, and predictor $\{E^{(0)}, \mathcal{G}^{(0)}, F^{(0)}\}$, initial discriminator $D^{(0)}$; a pre-defined classifier $\mathcal{Q}$; random noise $\mathcal{Z} \sim (0, I)$; the number of tasks $K$, the number of clients $N$, all tasks $\{\{T_i^k = \{X_i^k, Y_i^k\}\}_{i \in [N]}\}_{k \in [K]}$; server $S$; the number of rounds $\mathcal{T}$

**Parameter:** lambda $\lambda_d$, Laplace parameter $b = \frac{\Delta f}{\epsilon}$

**Output:** global models $\{E_g, \mathcal{G}_g, F_g\}$

1: $\{E_g^{(0)}, \mathcal{G}_g^{(0)}, F_g^{(0)}\} = \{E^{(0)}, \mathcal{G}^{(0)}, F^{(0)}\}$
2: **for** $k = 1, 2, \ldots, K$ **do**
3:    $\tau_G \sim Lap(b)$
4:    **for** $i = 1, 2, \ldots, N$ **do**
5:       Client $C_i$ trains $\mathcal{Q}$ on task $T_i^k$ for a few epochs and then uploads model updates $w_i^k$ to server $S$
6:    **end for**
7:    Server $S$ generates relational graph $G^k$ based on spatial and temporal attention mechanism and then broadcast $\{G_{i,:}^k\}_{i \in [N]}$ to each client respectively
8:    **for** $t = (k-1)\frac{\mathcal{T}}{K} + 1, (k-1)\frac{\mathcal{T}}{K} + 2, \ldots, (k-1)\frac{\mathcal{T}}{K} + \frac{\mathcal{T}}{K}$ **do**
9:       **for** $i = 1, 2, \ldots, N$ **do**
10:          $\{E_i^{(t-1)}, \mathcal{G}_i^{(t-1)}, F_i^{(t-1)}\} = \{E_g^{(t-1)}, \mathcal{G}_g^{(t-1)}, F_g^{(t-1)}\}$
11:          **if** k=1 **then**
12:             $\mathcal{E}_i^{(t)} = E_i^{(t-1)}(X_i^k, G_{i,:}^k)$
13:          **else**
14:             $\mathcal{E}_i^{(t)} = E_i^{(t-1)}(X_i^k, G_{i,:}^k) \cup \{\mathcal{G}_i^{(t-1)}(\mathcal{Z}, Y_i^j, G_{i,:}^j)\}_{j=1}^{k-1}$
15:          **end if**
16:          $\tilde{\mathcal{E}}_i^{(t)} = \mathcal{E}_i^{(t)} + \tau_i^{(t)}, \tau_i^{(t)} \sim Lap(b)$
17:          $\tilde{G}_{i,:}^k = G_{i,:}^k + \tau_G$
18:          $C_i$ upload $\{\tilde{\mathcal{E}}_i^{(t)}, \tilde{G}_{i,:}^k\}$ to server $S$
19:       **end for**
20:       Server $S$ trains and obtains $D^{(t)}$ by the max function of Equation (1)
21:       Server $S$ sends $D^{(t)}$ back to all clients $\{C_i\}_{[i \in [N]}$
22:       **for** $i = 1, 2, \ldots, N$ **do**
23:          Client $C_i$ trains and obtains Encoder $E_i^{(t)}$, Generator $\mathcal{G}_i^{(t)}$, and Predictor $F_i^{(t)}$ by the min function of Equation (1)
24:          Client $C_i$ uploads Encoder $E_i^{(t)}$, Generator $\mathcal{G}_i^{(t)}$, and Predictor $F_i^{(t)}$ to Server $S$
25:       **end for**
26:       Server $S$ aggregates the models by $E_g^{(t)} = \frac{1}{N}\sum_{i=1}^N E_i^{(t)}, \mathcal{G}_g^{(t)} = \frac{1}{N}\sum_{i=1}^N \mathcal{G}_i^{(t)}$, and $F_g^{(t)} = \frac{1}{N}\sum_{i=1}^N F_i^{(t)}$. Then, broadcast $\{E_g^{(t)}, \mathcal{G}_g^{(t)}, F_g^{(t)}\}$
27:    **end for**
28: **end for**
29: **return** $\{E_g^{(\mathcal{T})}, \mathcal{G}_g^{(\mathcal{T})}, F_g^{(\mathcal{T})}\}$

---

*Table 10.* FCL setup of GFedCL used in all training datasets.

| Description | | EMNIST-Letters | CIFAR-100 | TinyImageNet | US-States | TCGA-BRCA |
|---|---|---|---|---|---|---|
| $N$ | # of clients | 8 | 10 | 10 | 7 | - |
| $K$ | # of tasks per client | 6 | 4 | 6 | 6 | - |
| $C$ | # of classes per task | 4 | 20 | 30 | 7 | - |

*Table 11.* Hyperparameter settings of GFedCL used in all training datasets. "General" represents the default value of the hyperparameter if there is no specification.

| Description | | General | EMNIST-Letters | CIFAR-100 | TinyImageNet | US-States |
|---|---|---|---|---|---|---|
| $\mathcal{T}$ | # of rounds | 20 | - | - | - | - |
| $epoch$ | Local epochs | 50 | - | - | - | - |
| $B$ | Local batch size | 256 | - | - | - | - |
| $lr$ | Local learning rate | 1e-4 | - | - | - | - |
| $\lambda$ | Balance term | 1 | - | - | - | |
| $\Delta f$ | Dataset sensitivity | 1 | - | - | - | |
| $\epsilon$ | Privacy budget | 1 | - | - | - | |

---

**Algorithm 2** Differential Privacy for GFedCL

---

**Input:** data $x$
**Parameter:** DP parameter $\epsilon$, sensitivity $\Delta f$
**Output:** noised data $\tilde{x}$

1: $b = \frac{\Delta f}{\epsilon}$
2: $\tilde{x} = x + \tau_x, \tau_x \sim Lap(b)$
3: **return** $\tilde{x}$

---

