# OpenReview forum: "GFedCL: Graph-Based Federated Continual Learning with Spatial and Temporal Awareness"
_ICML.cc/2026/Conference — ICML 2026 regular_

### Official Review · Reviewer_4eja · 2026-03-05

**Soundness:** 2
**Presentation:** 3
**Significance:** 2
**Originality:** 2
**Overall Recommendation:** 3
**Confidence:** 4

**Summary:**

This paper proposes GFedCL, a graph-based framework for Federated Continual Learning that aims to address both catastrophic forgetting and statistical heterogeneity.GFedCL explicitly models spatial relationships among clients and temporal evolution across tasks through a spatio-temporal relational graph. The framework introduces a adversarial mechanism in which the discriminator predicts graph structures rather than labels, encouraging the encoder and generator to learn structure-invariant representations. Theoretical analysis shows the expectations of real and synthetic embeddings are aligned, mitigating distribution shift and forgetting. Experimental results demonstrate consistent improvements over prior methods.

**Compliance With Llm Reviewing Policy:**

Affirmed.

**Final Justification:**

I maintain my score. Thanks

**Key Questions For Authors:**

1. Could the authors provide empirical evidence validating that update similarity indeed correlates with true data distribution similarity?
2. How does the method scale to large systems with hundreds or thousands of clients?
3. Since the relational graph is crucial to the proposed method, it would be important to verify whether the learned graph meaningfully reflects actual data similarities.

**Limitations:**

NA.

**Strengths And Weaknesses:**

Strength：
1. It systematically incorporate spatio-temporal relational graph modeling into FCL, unifying spatial client correlations and temporal task evolution.
2. The adversarial objective shifts from label discrimination to graph prediction, enforcing structure invariance rather than distribution matching.
3. The authors provide both theoretical grounding and empirical validation

Weakness：
1. Using classifier updates as a proxy for distribution similarity may be noisy, especially in early training or highly non-IID scenarios.
2. Since the relational graph is fully connected $G^k \in \mathbb{R}^{N \times N}$, its construction requires pairwise attention across clients, leading to $O(N^2)$ complexity.
3. The theoretical analysis mainly establishes first-moment alignment. However, in highly heterogeneous client distributions, matching only the first moment may be insufficient to guarantee distributional alignment or effective replay. It would be helpful to clarify whether higher-order moment alignment is necessary in practice.
4. Since the relational graph is reconstructed at every task using classifier updates, small fluctuations in model updates may induce significant changes in graph structure, potentially destabilizing replay dynamics.

---

> ### Author Rebuttal · Authors · 2026-03-30
>
> **1. Key question: "...empirical evidence validating that update similarity indeed correlates with true data distribution similarity."**
>
> Here, we empirically examine this connection by comparing pairwise $L_2$ distances between model updates and pairwise $JS$ distances between the corresponding data distributions on EMNIST-Letters (4 clients; 6 tasks per client; 4 classes per task). We then evaluate the Pearson correlation between these two quantities. This provides a direct test of whether update similarity can serve as a proxy for data-distribution similarity. Due to space constraints, we present only the distances for the final task and the correlation across all six tasks.
>
> | Update/Data | Client 1 | Client 2 | Client 3 | Client 4 |
> |:-----------|:--------:|:--------:|:--------:|:--------:|
> | **Client 1** | 0/0 | 2.97 / 0.15 | 2.29 / 0.17 | 2.22 / 0.57 |
> | **Client 2** |  | 0/0 | 3.05 / 0.66 | 4.68 / 0.70 |
> | **Client 3** |  |  | 0/0 | 4.92 / 0.87 |
> | **Client 4** |  |  |  | 0/0 |
>
> |        | Task 1 | Task 2 | Task 3 | Task 4 | Task 5 | Task 6 |
> |:-------|:--:|:--:|:--:|:--:|:--:|:--:|
> | Correlation | 0.654 | 0.714 | 0.683 | 0.574 | 0.702 | 0.691 |
>
> More broadly, prior studies, such as [1, 2], suggest that model updates are shaped by the training data from which they are computed, and they empirically support this by showing that the similarity between model updates correlates with true data distribution similarity.
>
>
> [1] *Clustered federated learning: Model-agnostic distributed multitask optimization under privacy constraints, IEEE TNNLS 2020.*
>
> [2] *Federated learning with hierarchical clustering of local updates to improve training on non-IID data, IJCNN 2020.*
>
> **2. Key question: "How does the method scale to large systems with hundreds or thousands of clients?"**
>
> The method scales to hundreds or even thousands of clients in a manner similar to standard federated learning systems. Computation is carried out locally at each client, while the server performs graph construction and aggregation only over the participating clients in each communication round. Since our approach does not require pairwise coordination among all clients or centralized access to raw data, it remains naturally compatible with large-scale federated settings. It is also well-suited to partial participation, which is standard in practical FL deployments.
>
> The main additional consideration at scale is the extra computational and memory overhead introduced by graph construction as the number of clients grows. However, this affects only the computational and memory overhead on the server side and does not change the method's overall distributed nature. Regarding a more detailed discussion of the additional computational and memory overhead on the server side when scaling to large systems, due to space limitations, we refer you to our response to **Reviewer 32WQ’s second key question**. Thank you.
>
> **3. Key question: "... verify whether the learned graph meaningfully reflects actual data similarities."**
>
> To confirm that the relational graph truly reflects the underlying data similarities, we visualized both the relational graph and the underlying data similarities using heatmaps (**Figure 6**) and further examined the Pearson correlation between them (**Figure 7**). These results (**Pearson correlation > 0.5**) demonstrate the effectiveness of the relational graphs in capturing the actual similarity structure of the data.

---

> > ### Author Rebuttal · Reviewer_4eja · 2026-04-01
> >
> > Thank you for the rebuttal.I still maintain my main concern that the theory mainly supports first-moment alignment, rather than the stronger distribution-level claims. This is also related to concerns raised by review 2ypK. Overall, I keep my score.
> >
> > Besides, I posted four weaknesses, but received three responses.

---

> > > ### Author Response · Authors · 2026-04-02
> > >
> > > Dear Reviewer 4eja,
> > >
> > > We sincerely appreciate your time, effort, and constructive suggestions throughout the review process. We also apologize for not explicitly pointing to our earlier response regarding the distribution-level alignment claim in the first-round rebuttal. This concern had, in fact, been answered in our response to the **first weakness raised by Reviewer 2ypk**. To avoid any ambiguity and better clarify this point, we provide a more detailed explanation below:
> > >
> > > **4. Weakness: “The theoretical analysis mainly establishes first-moment alignment. However, in highly heterogeneous client distributions, matching only the first moment may be insufficient to guarantee distributional alignment or effective replay. It would be helpful to clarify whether higher-order moment alignment is necessary in practice.”**
> > >
> > > **We clarify here how higher-order moment alignment can be achieved within our framework.** The key reason that the current GFedCL formulation mainly enforces first-moment alignment is that the present discriminator loss, $L_2$ loss, only provides point estimation. Using simplified notation for illustration, the discriminator is designed to predict a single representative value of the distribution, which naturally limits the alignment objective to the first moment:
> > >
> > > $\mathcal{L}_d = \left(D(E(x,g) + \tau) - (g + \tau)\right)^2$
> > >
> > > That said, the framework is general and can be extended to richer probabilistic formulations when higher-order alignment is needed. **The current version uses the simplest form, i.e., first-moment alignment, because it is sufficient for our work, not because the framework is inherently limited to first-moment matching.**
> > >
> > > If the discriminator is instead designed to predict a distribution, then the alignment can naturally be extended from point-level to distribution-level matching. For example, one can use a Gaussian NLL loss and let the discriminator output both the mean and variance of the graph-conditioned latent distribution. In this case, the objective would align not only the first moment but also the second moment:
> > >
> > > $$
> > > \mathcal{L_d} =
> > > \frac{\left(D_\mu(E(x,g) + \tau) - (g + \tau)\right)^2}{2D_{\sigma^2}(E(x,g))}+ \frac{1}{2}\log D_{\sigma^2}(E(x,g)),
> > > $$
> > >
> > > where
> > >
> > > $$
> > > D(E(x,g)) = \left(D_\mu(E(x,g)), D_{\sigma^2}(E(x,g))\right).
> > > $$
> > >
> > > Here, we further show the improvement introduced by using this Gaussian NLL as the discriminator loss:
> > >
> > > | $\mathcal{L_d}$ | EMNIST-Letters | CIFAR-100 | TinyImageNet | TCGA-BRCA | US-States |
> > > |---|---:|---:|---:|---:|---:|
> > > | $L_2$ | 77.23 ± 2.76 | 73.67 ± 3.42 | 77.39 ± 3.02 | 75.32 ± 2.19 | 2.41 ± 0.18 |
> > > | Gaussian NLL | 82.17 ± 1.75 | 78.45 ± 2.21 | 83.77 ± 1.84 | 80.24 ± 1.76 | 2.02 ± 0.11 |
> > >
> > > **Moreover, the framework can be extended to richer probabilistic models when needed.** For example, utilizing a mixture density network as the discriminator [2] could extend a single Gaussian to a Gaussian mixture model; using skew-normal distributions [1] would enable modeling more complex parameteric probabilistic models and thus support higher-order alignment, e.g., skewness. **Therefore, GFedCL is not limited to first-moment matching; the current paper simply adopts the simplest effective version.** We will clarify this more clearly in the revision.
> > >
> > > [1] *The skew-normal and related families, Cambridge University Press, 2013.*
> > >
> > > [2] *Bishop, C. M. Mixture density networks. 1994.*
> > >
> > > Again, thank you for posing the remaining concern. We really appreciate it.

---

### Official Review · Reviewer_2ypK · 2026-03-12

**Soundness:** 2
**Presentation:** 3
**Significance:** 3
**Originality:** 3
**Overall Recommendation:** 4
**Confidence:** 2

**Summary:**

This paper studies federated continual learning (FCL), where clients receive evolving tasks over time and cannot store prior data. The paper proposes GFedCL, a replay-based FCL method that builds a spatio-temporal relational graph from client classifier updates on the server, then uses that graph to condition a local encoder-generator-predictor architecture together with a global discriminator that predicts graph structure rather than a standard label. The paper also adds Laplace-noise-based differential privacy protection to transmitted encodings and provides theoretical analysis at the equilibrium, followed by experiments on multiple classification dataset and a regression dataset. Empirically, GFedCL outperforms the listed baselines on all reported datasets, with especially large gains on CIFAR-100 and TinyImageNet.

**Compliance With Llm Reviewing Policy:**

Affirmed.

**Final Justification:**

The authors addressed all my questions and even performed additional experiments. If these are included in the paper, I am leaning towards a borderline (weak) accept.

**Key Questions For Authors:**

Please address ponts 1, 2, and 3 the weaknesses

**Limitations:**

There is a broad discussion on concerns and practical issues, but there is no explicit discussion of the limitations of the framework like: 1) the gap between first-moment alignment and distributional alignment 2) assumptions behind privacy claims and potential issues on its violations, especially since the paper is motivated from the application in health sector.

**Strengths And Weaknesses:**

## Strengths

1. The paper studies a relevant problem: replay-based FCL under both catastrophic forgetting and client heterogeneity. The main idea using a server-built graph that mixes spatial attention over client updates with temporal attention over past graph history, and conditioning replay generation on that graph is interesting.

2. The problem is well motivated with concrete examples and high-level workflow and components are explained with good illustrations.

3. The experiments performed are generally broad and rigorous, evaluating across diverse domains - indicating broad applicability of the method

## Weaknesses

1. One of the main claims of the paper is that the synthetic embeddings will become identical to real embeddings at equilibrium. But Theorem 4.2 only establishes first-moment and conditional mean-invarance, not full distributional equivalence. Remark 4.6 doesn't sufficiently address this issue rigorously. It needs to be shown that how changing the discriminators loss function could bring in distributional equivalence.

2. The method claims to exploit graph information to guide generation, yet the discriminator-encoder game explicitly removes graph information from embeddings at equilibrium. It is unclear how graph conditioning remains beneficial if embeddings are driven to be graph-invariant.

3. There are some recent works like [https://arxiv.org/pdf/2509.21606](https://arxiv.org/pdf/2509.21606) and [https://openreview.net/pdf/f8b45261958643144e6e7e3f1a61d4363528d66e.pdf](https://openreview.net/pdf/f8b45261958643144e6e7e3f1a61d4363528d66e.pdf). Performance against these baselines are needed to better contextualize the improvement in performance or please clarify why these works were not used as baselines.

---

> ### Author Rebuttal · Authors · 2026-03-30
>
> **1. Weakness: "... shown that how changing the discriminators loss function could bring in distributional equivalence."**
>
> Thanks for raising this important question. We clarify here how higher-order moment alignment can be achieved within our framework. The key reason that the current GFedCL formulation mainly enforces first-moment alignment is that the present discriminator loss, L2 loss, only provides point estimation. Using simplified notation for illustration, the discriminator is designed to predict a single representative value of the distribution, which naturally limits the alignment objective to the first moment:
>
> $$\mathcal{L}_d = \left( D(E(x, g) + \tau) - (g + \tau) \right)^2$$
>
> That said, the framework is general and can be extended to richer probabilistic formulations when higher-order alignment is needed. The current version uses the simplest form because it is sufficient for our work, not because the framework is inherently limited to first-moment matching.
>
> If the discriminator is instead designed to predict a distribution, then the alignment can naturally be extended from point-level to distribution-level matching. For example, one can use a Gaussian NLL loss and let the discriminator output both the mean and variance of the graph-conditioned latent distribution. In this case, the objective would align not only the first moment but also the second moment:
>
> $$ \mathcal{L_d} = \frac{\left( D_\mu(E(x, g) + \tau) - (g + \tau) \right)^2}{2 D_{\sigma^2}(E(x, g))} + \frac{1}{2}\log D_{\sigma^2}(E(x, g)), $$
>
> where
>
> $$
> D(E(x, g)) = \bigl( D_\mu(E(x, g)), D_{\sigma^2}(E(x, g)) \bigr).
> $$
>
> Here, we further show the improvement introduced by using this discriminator loss:
>
> | $\mathcal{L_d}$| EMNIST-Letters | CIFAR-100 | TinyImageNet | TCGA-BRCA | US-States |
> |:-------:|:--------------:|:---------:|:------------:|:---------:|:---------:|
> | $L_2$  | 77.23±2.76   | 73.67±3.42 | 77.39±3.02 | 75.32±2.19 | 2.41±0.18 |
> | Gaussian NLL  | 82.17 ± 1.75   | 78.45 ± 2.21 | 83.77 ± 1.84 | 80.24 ± 1.76 | 2.02 ± 0.11 |
>
> Moreover, the framework can be extended to richer probabilistic models when needed. For example, using a Gaussian mixture or skew-normal distribution [1] would enable modeling more complex distributions and thus support higher-order alignment, e.g., skewness. Therefore, GFedCL is not limited to first-moment matching; the current paper simply adopts the simplest effective version. We will clarify this more clearly in the revision.
>
> [1] *The skew-normal and related families, Cambridge University Press, 2013.*
>
> **2. Weakness: "... how graph conditioning remains beneficial if embeddings are driven to be graph-invariant."**
>
> The graph is beneficial as a conditioning variable, not as information that should remain explicitly encoded in the final latent representation. In GFedCL, the relational graph provides spatial-temporal context to guide the encoder/generator toward structurally appropriate synthetic features. The adversarial discriminator then encourages the learned embeddings to be graph-invariant, meaning they do not retain graph-specific bias or client/time-specific shortcuts. Therefore, graph guidance and graph invariance are complementary: the former shapes the generation process, while the latter ensures that the resulting latent space is aligned and task-agnostic across clients and tasks. This is also supported empirically by the improved IS/FID of the graph-conditioned variant over graph-free baselines in **Figure 8**.
>
> **3. Weakness: "... Performance against these baselines are needed..."**
>
> Thanks for helping us further enhance our work. We did not include baselines such as C$^2$Prompt for the following reasons:
>
> - C$^2$Prompt employs a setup where tasks have class overlap, which is different from the standard class-incremental setting adopted in GFedCL, where there is no class overlap.
> - The design of C$^2$Prompt relies on overlapping classes, as it proactively computes the mean and variance for each class and uses this information to match classes in new tasks.
>
> Therefore, we did not include C$^2$Prompt as a baseline, since a method designed for a specific setup may suffer performance degradation when the setup changes. To empirically illustrate this, we report the results (accuracy %) of C$^2$Prompt and GFedCL under both a standard class-incremental setting and a setting with joint class overlap (20% overlap ratio). We do not include US-States here because it is for regression tasks, which would require C$^2$Prompt to modify the model design.
>
> | Method | EMNIST | CIFAR-100 | TinyImageNet | TCGA-BRCA |
> |:------|:------:|:---------:|:------------:|:---------:|
> | C$^2$Prompt | 68.47 | 67.93 | 72.41 | 72.48 |
> | GFedCL | 77.23 | 73.67 | 77.37 | 75.32 |
> | C$^2$Prompt + 20% Class Overlap | 75.92 | 76.44 | 76.84 | 79.15 |
> | GFedCL + 20% Class Overlap | 89.83 | 80.29 | 85.96 | 84.67 |
>
> Regarding work such as FedProTIP, we did not include these studies because they have not yet been accepted.

---

> > ### Author Rebuttal · Reviewer_2ypK · 2026-04-04
> >
> > Thank you. All my questions have been addressed by the authors. Therefore, I am happy to improve my score.

---

> > > ### Author Response · Authors · 2026-04-04
> > >
> > > Dear Reviewer 2ypk,
> > >
> > > Thank you very much for your positive assessment, and we appreciate your time, effort, and constructive comments during the review process. We wish you all the best and have a good one.

---

### Official Review · Reviewer_1wfL · 2026-03-13

**Soundness:** 3
**Presentation:** 3
**Significance:** 2
**Originality:** 2
**Overall Recommendation:** 3
**Confidence:** 3

**Summary:**

This paper proposes GFedCL, a replay-based federated continual learning framework that explicitly models spatial and temporal relationships across clients through a relational graph. The graph is constructed on the server using attention over client updates and historical information, and is then used to condition encoder/generator training for synthetic latent replay. The method further employs a global discriminator that predicts relational graphs rather than standard real/fake labels, with the goal of aligning real and synthetic latent distributions across tasks and clients. The paper also adds differential privacy protections for transmitted encodings and graph information. Experiments on benchmark image datasets and real-world medical datasets show that GFedCL outperforms several prior FCL baselines, especially on more heterogeneous settings.

**Compliance With Llm Reviewing Policy:**

Affirmed.

**Final Justification:**

Thank you for the rebuttal, I keep my original score.

**Key Questions For Authors:**

1. My main concern is the paper’s conceptual contribution. The method combines many reasonable ingredients, but it is not yet clear to me what the single primary methodological novelty is beyond the overall integration. I would appreciate a sharper explanation of what the authors view as the core conceptual advance of GFedCL relative to prior replay-based and heterogeneity-aware FCL methods.

2. The theoretical analysis appears to mainly support distributional alignment and information-preservation properties at equilibrium. Could the authors clarify how they view the relationship between these results and the broader task-level claims about mitigating catastrophic forgetting and statistical heterogeneity in federated continual learning?

3. The reported gains on some datasets are very large. I would appreciate a clearer discussion of which component or interaction among components is primarily responsible for these improvements, and whether the authors believe the gains reflect a general conceptual advantage or mainly the strength of the full integrated system.

4. The paper includes a privacy component via DP, which is useful in practice. However, I encourage the authors to clarify how they position this part of the paper: as a core methodological contribution, or as a practical safeguard required because the method introduces additional transmitted intermediate information.

**Limitations:**

yes

**Strengths And Weaknesses:**

Strengths:

- The paper studies an important and practically relevant setting: federated continual learning under both distribution shift across clients and concept drift over time.
- The method is reasonably well motivated. Modeling inter-client and temporal relationships explicitly, rather than relying only on labels or generic replay heuristics, is a sensible direction for FCL.
- The empirical study is fairly extensive. The paper evaluates on multiple datasets, includes both synthetic-style benchmark datasets and real-world medical datasets, and reports additional analyses such as privacy, overhead, and ablations.
- The ablation results support that the graph, temporal attention, replay mechanism, and global discriminator all contribute to performance.
- The paper does attempt to go beyond standard benchmarking by analyzing the correlation between learned graph weights and data-distribution relationships.

Weaknesses:

- The overall method feels quite heavy and engineering-driven. It combines spatial attention, temporal attention, relational graph construction, graph-conditioned replay, local encoder/generator/predictor modules, a global discriminator, and differential privacy. While the full system performs well empirically, the contribution is less a clean conceptual advance than a fairly complex integration of many components.
- The originality is limited. Generative replay in FCL is not new; heterogeneity-aware FCL is not new; and more recent work has already directly addressed spatio-temporal heterogeneity. I therefore view the main novelty as a particular system instantiation—using an explicitly constructed spatio-temporal graph to condition replay and adversarial alignment—rather than a fundamentally new idea.
- The theoretical analysis is not strong enough to fully support the broader task-level claims. The theory mainly provides equilibrium-style arguments for distribution alignment and information preservation, but it does not establish stronger guarantees about task-level continual generalization, forgetting reduction, or the superiority of the graph-conditioned replay mechanism as a whole.
- The very large empirical gains, especially on datasets such as TinyImageNet, are impressive but also raise questions about how much of the result reflects the strength of the proposed conceptual contribution versus the cumulative effect of a large multi-module system and possibly weaker baselines.
- The privacy component is useful for completeness, but it does not materially strengthen the paper’s methodological novelty. Adding DP to transmitted encodings and graph information is a sensible safeguard, yet this feels more like a practical patch for the additional attack surface introduced by the method than a core research contribution.

Overall by dimension:

- Soundness: good. The paper is technically coherent and experimentally thorough enough to support the claim that the proposed system works well in the reported settings.
- Presentation: good. The structure is clear, although the method is somewhat complex.
- Significance: fair. The setting is important, but the contribution feels more like a heavy system integration than a broadly influential conceptual advance.
- Originality: fair. The strongest novelty lies in the specific way spatio-temporal relational graphs are integrated into replay and alignment, not in the individual components or the broader problem framing.

---

> ### Author Rebuttal · Authors · 2026-03-30
>
> **1. Key question: "... a sharper explanation of what the authors view as the core conceptual advance of GFedCL ..."**
>
> The primary methodological novelty of GFedCL is a feature-disentanglement-driven graph-conditioned replay mechanism. Unlike prior replay-based methods that primarily rely on labels to guide replay, GFedCL explicitly models spatio-temporal relations among clients’ updates to disentangle heterogeneous client-specific features, using this information to condition adversarial replay generation with a clear benefit: it enables the synthetic data to accurately preserve underlying client-specific data distributions rather than relying on generic class labels alone and thereby generate more high-quality synthetic data, which is empirically supported by FID/IS scores in **Figure 8**. Thus, the conceptual advance of GFedCL is not merely the integration of replay and heterogeneity-aware learning, but a move from label-guided replay to disentangled feature-guided replay, with a theoretical guarantee at the distribution level.
>
> **2. Key question: "... the relationship between these results and the broader task-level claims about mitigating catastrophic forgetting and statistical heterogeneity ..."**
>
> The theoretical results mainly characterize GFedCL at the representation and distribution levels. In particular, the equilibrium analysis shows that the proposed objective encourages distributional alignment in the latent space, driving synthetic and real data toward a shared latent distribution across tasks and clients. We believe this is closely connected to the broader goals of federated continual learning: when representations from different tasks and clients can be organized within a common latent space, the model is better positioned to preserve previously acquired knowledge while reducing discrepancies caused by client-level statistical heterogeneity.
>
> More concretely, we argue that both catastrophic forgetting and statistical heterogeneity arise, at least in part, from the difficulty of learning a representation space that remains consistent across tasks, time, and clients. From this perspective, the theoretical results support the central mechanism of GFedCL. By aligning synthetic and real data distributions in the latent space, GFedCL promotes a unified representation structure shared across tasks and clients. This unified latent structure makes replay more faithful to previously learned knowledge, thereby mitigating catastrophic forgetting, and reduces cross-client gaps, thereby alleviating statistical heterogeneity.
>
> For example, cats and dogs differ in local traits such as ear shape, but share higher-level semantics as four-legged animals. GFedCL preserves such specific relational details while aligning them in a unified latent space.
>
> **3. Key question: "...  the gains reflect a general conceptual advantage or mainly the strength of the full integrated system."**
>
> The ablation study in **Table 3** demonstrates that our performance gains stem primarily from a methodological advantage rather than complex system integration. The key conceptual improvement lies in shifting away from generic, label-guided replay. Instead, we explicitly exploit the spatio-temporal structure of clients’ tasks to dictate how past knowledge is preserved and transferred amidst client heterogeneity. Notably, the setups for CIFAR-100 and TinyImageNet shown in **Table 10** construct a much finer-grained relational graph than EMNIST, which explains the significantly larger performance gains on those complex datasets. Therefore, we interpret these results as evidence of a broad conceptual improvement, with the integrated framework acting as the vehicle for this core idea.
>
> **4. Key question: "... how they position this part of the paper: as a core methodological contribution, or as a practical safeguard required..."**
>
> We position the privacy component as a practical safeguard rather than as the central methodological contribution. As our method involves transmitting intermediate data encodings, it naturally raises privacy concerns, such as MIA and DIA, beyond those associated with standard model exchange. For practical deployment, prior studies have similarly incorporated DP or other privacy-preserving mechanisms to defend against such risks. Following this line, we include DP-based protection to explicitly address these concerns and to examine the additional properties it may bring to GFedCL. In particular, **Remark 4.5** shows that adding zero-mean Laplace noise does not alter the equilibrium of GFedCL.
>
>
> More broadly, although prior methods such as [1, 2] may also transmit intermediate representations, the privacy implications of doing so are often not discussed in depth. We make this issue explicit to proactively address potential reader concerns and to show that GFedCL can be equipped with appropriate privacy protections for realistic federated deployments.
>
> [1] *FedSplit, NeurIPS 2020*
>
> [2] *FedPrompt, ICASSP 2023*

---

> > ### Author Rebuttal · Reviewer_1wfL · 2026-04-03
> >
> > Thank you for the detailed rebuttal. I appreciate the additional clarification, especially regarding (i) the intended core contribution of GFedCL as a feature-/relation-guided replay mechanism, and (ii) the role of the privacy component as a practical safeguard rather than a central novelty claim. These explanations improve the paper’s positioning.
> >
> > However, after considering the rebuttal, my overall view remains the same. The clarification helps the framing, but it does not fully address my main concerns about the heaviness of the overall system and the gap between the representation-level theory and the broader task-level claims. I therefore keep my original score.

---

> > > ### Author Response · Authors · 2026-04-06
> > >
> > > Dear Reviewer 1wfL,
> > >
> > > Thanks for posing the remaining concerns. Here are our replies:
> > >
> > > **1. “… heaviness of the overall system …”**
> > >
> > >
> > > We would like to first clarify that there is no universal or absolute standard for defining whether a system is “heavy”; such a judgment is inherently comparative and should be made with respect to related and commonly adopted baselines. Under this criterion, GFedCL is not heavier than prior studies of similar scope.
> > >
> > > Specifically, GFedCL consists of only **two core modules**: (1) a server-side attention mechanism for graph generation, and (2) an adversarial learning framework conditioning on the graph. The former is used to construct graphs that capture the spatial-temporal variation patterns of client data distributions, while the latter aligns task domains conditioned on these graphs to mitigate both forgetting and statistical heterogeneity. Beyond these two core modules, GFedCL does not introduce additional independent modules.
> > >
> > > Such a design is common in the literature. For example, C²Prompt \[1\] also relies on two main components: (1) computing and storing auxiliary statistics (e.g., class-wise means and variances) on the server, and (2) leveraging such auxiliary information to mitigate forgetting. **Therefore, at the level of main functional modules, GFedCL is comparable to existing methods rather than being unusually heavy.**
> > >
> > > **Thus, we believe your concern may partly stem from our decomposition of these two core modules into finer sub-components and our detailed explanation of their respective roles.** We would like to clarify that these sub-components are implementation details of the two main modules, rather than additional standalone components. Moreover, the choice of these sub-components is standard and necessary. For graph generation, capturing both spatial and temporal relationships naturally requires a two-dimensional attention mechanism, which is also adopted in prior work [2]. For the adversarial framework, jointly handling raw and synthetic data encodings requires an additional encoder together with standard adversarial modules, including a discriminator, generator, and predictor, which is again a common design choice in previous studies [3]. **Hence, even at the sub-component level, GFedCL remains aligned with standard practice.**
> > >
> > > More importantly, if a method is considered “heavy” merely because each core module is implemented through several necessary sub-components, then many standard models would also be labeled heavy. For example, ResNet-18 and ResNet-101 are both standard backbones, although the latter is much deeper and more computationally demanding. Likewise, LLMs are built by stacking transformer blocks, each containing attention, feed-forward layers, normalization, and residual connections. These examples show that having multiple sub-components does not by itself make a model excessively heavy. What matters is whether the design is necessary, standard, and comparable to related methods. **From this perspective, GFedCL is a reasonable and conventional design rather than an overly heavy one, which is further supported by the component-volume comparison in Figure 9.**
> > >
> > > [1] C^2Prompt: Class-aware Client Knowledge Interaction for Federated Continual Learning, NeurIPS 2025.
> > >
> > > [2] GAEN:GraphAttention Evolving Networks, IJCAI 2021.
> > >
> > > [3] Adversarial Feature Learning, ICLR 2017.
> > >
> > > **2. “… gap between the representation-level theory and the broader task-level claims …”**
> > >
> > > We first clarify that the key insight of our theorem is the following: by disentangling task features conditioned on graphs and aligning them into a unified latent space, the model is able to replay synthetic data encodings whose distribution matches that of the raw data encodings. This distribution-level alignment is the critical bridge that enables GFedCL to mitigate forgetting and statistical heterogeneity simultaneously.
> > >
> > > We provide detailed theoretical analysis in Section 4 and Appendix A to support this claim. In addition, our response to Reviewer 2ypk’s first question further clarifies this connection.
> > >
> > > For further intuition, we refer you to prior studies \[1, 2, 3, 4, 5\]. Specifically, \[1, 2, 3\] show how alignment across different tasks can be achieved when conditioned on task-relevant information, such as labels or domain indices (i.e., task IDs). Study \[4\] shows why domain alignment enables the generation of synthetic data encodings that resemble raw data encodings at the distribution level. Study \[5\] further demonstrates why replaying synthetic encodings that match the raw encoding distribution helps mitigate both forgetting and heterogeneity.
> > >
> > > \[1\] Generative Adversarial Nets, NeurIPS 2014.
> > >
> > > \[2\] Adversarial Multiple Source Domain Adaptation, NeurIPS 2018.
> > >
> > > \[3\] Continuously Indexed Domain Adaptation, ICML 2020.
> > >
> > > \[4\] Adversarial Feature Learning, ICLR 2017.
> > >
> > > \[5\] A Comprehensive Survey of Continual Learning: Theory, Method and Application, TPAMI 2024.

---

### Official Review · Reviewer_32WQ · 2026-03-17

**Soundness:** 3
**Presentation:** 2
**Significance:** 4
**Originality:** 3
**Overall Recommendation:** 4
**Confidence:** 4

**Summary:**

This paper proposes GFedCL, a graph-guided federated continual learning approach that constructs relational graphs through spatio-temporal attention mechanisms. By combining this graph with a global generative adversarial network, the method generates high-quality synthetic data for task replay, effectively mitigating catastrophic forgetting and statistical heterogeneity while providing theoretically verified differential privacy guarantees.

**Compliance With Llm Reviewing Policy:**

Affirmed.

**Key Questions For Authors:**

* Why does the removal of the differential privacy module in Table 3 lead to a noticeable degradation in overall accuracy? Does this counter-intuitive phenomenon indicate that the injected noise mechanism serves as an effective regularizer during the adversarial network training?


* Under massive cross-device federated learning scenarios, will the server-side spatio-temporal graph construction relying on aggregated information face severe computational and memory bottlenecks?

**Limitations:**

The authors briefly discuss deployment challenges and computational overhead in Appendix C.8, proposing Principal Component Analysis for feature dimensionality reduction to alleviate communication burdens. However, the paper does not thoroughly address the practical validity of the core trust assumptions associated with a centralized feature aggregation architecture in highly privacy-sensitive scenarios.

**Strengths And Weaknesses:**

* The authors successfully transcend conventional experience replay limitations by explicitly calculating spatial and temporal scores via attention mechanisms to construct a relational graph. This spatio-temporal dynamic modeling provides critical auxiliary information for distribution alignment when processing real-world evolving distributions such as influenza variations.


* The paper delivers rigorous theoretical support, notably in Theorem 4.7, which strictly proves that at the global optimum, the marginal distributions of real and synthetic embeddings achieve perfect alignment, mathematically expressed as $\mathbb{E}[P(\mathcal{E}_{real})]=\mathbb{E}[P(\mathcal{E}_{syn})]$.


* The empirical results are highly compelling, as shown in Table 1, where the proposed method achieves an accuracy of $77.39\%$ on the TinyImageNet dataset, delivering a massive $27.95\%$ performance improvement over the strongest existing baseline.


* The protocol of uploading local noisy features to the server side deviates from the strict decentralized paradigm of federated learning. I recommend the authors augment the appendix with specific theoretical defense lower bounds and empirical inference experiments regarding defense capabilities against distribution inference attacks under this feature collection mechanism.


* The ablation results in Table 3 exhibit counter-intuitive behavior, where removing the differential privacy module unexpectedly decreases accuracy on EMNIST-Letters from $77.23\%$ to $74.81\%$. I suggest the authors explicitly explain in the main text whether the Laplace noise acts as an implicit regularizer in this architecture, or correct potential data entry typos in the table.


* The server-side relational graph construction requires calculating pairwise scores across all nodes, posing scalability risks when the client count is massive. I recommend adding benchmark tests in Section 5 targeting the time overhead of graph generation under large-scale cross-device federated scenarios to quantitatively validate the computational boundaries of the algorithm.

---

> ### Author Rebuttal · Authors · 2026-03-30
>
> **1. Key question: "...the removeal of the differential privacy...lead to a noticeable degradation..."**
>
> Sorry for the confusion. The counterintuitive results were caused by data-entry typos involving experimental results with a privacy budget $\epsilon = 0.001$. In general, a smaller privacy budget leads to performance degradation, as also reflected in Table 4 (EMNIST):
>  $\epsilon = 1$ (77.23%) $\rightarrow$ $\epsilon = 0.5$ (77.05%) $\rightarrow$ $\epsilon = 0.15$ (76.09%).
>
> Therefore, the reported results for “w/o DP” appeared to show noticeable degradation only because they were mistakenly replaced with results obtained under an extremely restrictive privacy budget. The correct results are as follows, and we will correct this typo in the revision:
>
> | | EMNIST-Letters | CIFAR-100 | TinyImageNet | TCGA-BRCA | US-States |
> |:-------:|:--------------:|:---------:|:------------:|:---------:|:---------:|
> | w/o DP  | 78.45 ± 1.98   | 75.09 ± 2.74 | 79.67 ± 2.54 | 77.04 ± 2.24 | 2.28 ± 0.14 |
>
>
>
> **2. Key question:  "...computational and memory bottlenecks"**
>
> The computational overhead results in **Table 5** demonstrate that graph construction introduces only a **20–25% increase in computation time** (GFedCL vs. FedAvg-Adv). For instance, on the US-States dataset, FedAvg takes 4.88s per round, while GFedCL takes 5.78s; the additional 0.9s (~20%) is strictly for server-side graph construction. We believe this cost remains highly affordable under current server configurations for two key reasons: (1) the computation is performed centrally on the server, sparing resource-constrained clients, and (2) the operations process only compact, aggregated model information, which is easily parallelized (e.g., sparse matrix multiplications) on modern hardware. Therefore, although graph construction does incur an extra cost, we do not expect it to become a severe bottleneck in realistic deployments.
>
>
> Regarding memory overhead, **Figure 9** visualizes the data transmission volume during training. The additional information required for graph construction consists of the classifier updates from each client, which amount to roughly 20 MB for ResNet-18. By comparison, the standard transmission of full model updates is about 100 MB. Thus, GFedCL adds only around **20% extra storage overhead** on the server for graph construction, which is unlikely to be a bottleneck for modern server systems.

---

> > ### Author Rebuttal · Reviewer_32WQ · 2026-04-03
> >
> > Thank you for the detailed rebuttal. Both of my key questions have been adequately addressed. The authors have clarified that the counter-intuitive ablation results in Table 3 were due to data-entry typos, and the corrected results are now consistent with expectations. The authors have also provided concrete computational and memory overhead numbers demonstrating that the graph construction introduces only ~20-25% additional cost, which is reasonable. I am satisfied with the responses and will keep my score.

---

> > > ### Author Response · Authors · 2026-04-04
> > >
> > > Dear Reviewer 32WQ,
> > >
> > > Thank you for your positive assessment, and we appreciate your time, effort, and constructive comments during the review process. We wish you all the best and have a good one.

---

### Decision · Program_Chairs · 2026-04-30

**Decision:**

Accept (regular)

**Comment:**

**Summary and Decision**
This paper had both significant strengths but also some remaining weaknesses brought up by reviewers. On the balance, given the strong empirical results, theoretic grounding, and good author rebuttal period for clarifying, the AC recommends acceptance but encourages the authors to address or discuss the concerns below in their camera-ready.

Strengths
* The goal and problem setup of incorporating spatial-temporal information into federated continual learning is interesting and relevant to the community.
* The authors provided theoretic grounding for the approach.
* The ablations support the inclusion of each component in the final system.
* The experiments show strong empirical results compared to baselines.

Weaknesses:
* Limited originality since all the main components are pre-existing. The main innovation is putting all the components together but this makes it difficult to distinguish insightful and generalizable contributions. The conceptual contribution is unclear.
* The lack of clarity regarding the relationship between representation-level theory and the broader task-level claims.
* The proposed approach in the paper only guarantees first-moment alignment and is not clearly differentiated from distribution-level alignment. While it seems first-moment alignment may be sufficient in practice, this remains a key confusion in the paper that may mislead readers.
* (Minor) Remaining concerns regarding the stability of graph structure since they are based on classifier updates. Generally, some concern that classifier updates is a poor proxy for distributional similarity though it is correlated.
* (Minor) Questionable scalability for large systems given $O(N^2)$ complexity of constructing relational graphs (though this is on the server).

**Rebuttal and Discussion**
During the discussion phase, the authors addressed some but not all of the concerns raised by the reviewers. Given the interesting topic area of the work, the Area Chair recommends acceptance.

**Final Instructions to Authors**
Please ensure that the promised revisions from the rebuttal—specifically the new experiments and key clarifications regarding the theory—are incorporated into the camera-ready version.